

# Modeling boreal forest's mineral soil and peat C stock dynamics with Yasso07 model coupled with updated moisture modifier

Boris Ťupek[1], Aleksi Lehtonen[1], Alla Yurova[2], Rose Abramoff[3,5,9], Stefano Manzoni[4], Bertrand Guenet[5], Elisa Bruni[5], Samuli Launiainen[1], Mikko Peltoniemi[1], Shoji Hashimoto[6], Xianglin Tian[7,8], Juha Heikkinen[1], Kari Minkkinen[7], and Raisa Mäkipää[1]

[1]Natural Resources Institute Finland (LUKE), Helsinki, 00790, Finland

[2]Northwest Institute of Eco-environment and Resources, Lanzhou, 730000, China

[3] Lawrence Berkley National Laboratory, University of California, Berkeley, 94720, USA

[4]Stockholm University, Stockholm, 10691, Sweden

[5]Laboratoire de Géologie, L'École Normale Supérieure (ENS), Paris, 75005, France

[6]Forestry and Forest Products Research Institute (FFPRI), Tsukuba, 305-8687, Japan

[7]Helsinki University, Helsinki, 00014, Finland

[8]College of Forestry, Northwest A & F University, Shaanxi, 712100, China

[9]Ronin Institute, Montclair, New Jersey, 07043-2314, USA

Corresponding author: Boris Ťupek (boris.tupek@luke.fi)

Keywords: soil water content (SWC), heterotrophic respiration, $CO_2$ emissions, soil organic carbon (SOC), Yasso07 model, environmental modifier, boreal forest

## Highlights

- The revision of functional water control in soil C and Earth systems models with large data across scales is needed to improve their projections.
- Substituting the Yasso07 soil C model's original dependency of decomposition on precipitation with a moisture function improved modelled SOC stocks and $CO_2$ emissions in a boreal forest catena of mineral and organic/peat soils.
- The moisture function calibrated with Bayesian MCMC SOC and $CO_2$ data assimilation showed an optimum of decomposition in dry well-drained soils.
- Using forest-mire SOC and $CO_2$ data together in model optimization was crucial to account for a spatial moisture gradient and its temporal variation in long- and short-term C dynamics.





**Abstract**

As soil microbial respiration is the major component of land $CO_2$ emissions, differences in the functional dependence of

respiration on soil moisture among the Earth system models (ESM) contributes significantly to the uncertainties in their

projections.

Using soil organic C (SOC) stocks and $CO_2$ data from a boreal forest – mire ecotone in Finland and Bayesian data assimilation,

we revised the precipitation-based environmental function of the Yasso07 soil carbon model. We fit this function to the

observed microbial respiration response to moisture and compared its performance against the original Yasso07 model and the

version used in the JSBACH land surface model with a reduction constant for decomposition rates in wetlands.

The Yasso07 soil C model coupled with the calibrated unimodal moisture function with an optimum in dry soils accurately

reconstructed observed SOC stocks and soil $CO_2$ emissions and clearly outperformed previous model versions on paludified

organo-mineral soils in forested peatlands and water-saturated organic soils in mires. The best estimate of the posterior

moisture response of decomposition used both measurements of SOC stocks and $CO_2$ data from the full range of moisture

conditions (from dry/xeric to wet/water-saturated soils). We observed unbiased residuals of SOC and $CO_2$ data modelled with

the moisture optimum in well-drained soils, suggesting that this modified function accounts more precisely for the long-term

SOC change dependency according to ecosystem properties as well as the contribution of short term $CO_2$ responses including

extreme events.

The optimum moisture for decomposition in boreal forests was in dry well-drained soils instead of the mid-range between dry

and water-saturated conditions as is commonly assumed among many soil C and ESM models. Although the unimodal moisture

modifier with an optimum in well-drained soils implicitly incorporates robust biogeochemical mechanisms of SOC

accumulation and $CO_2$ emissions, it needs further evaluation with large scale data to determine if its use in land surface models

will decrease the uncertainty of projections.





**Graphical abstract**

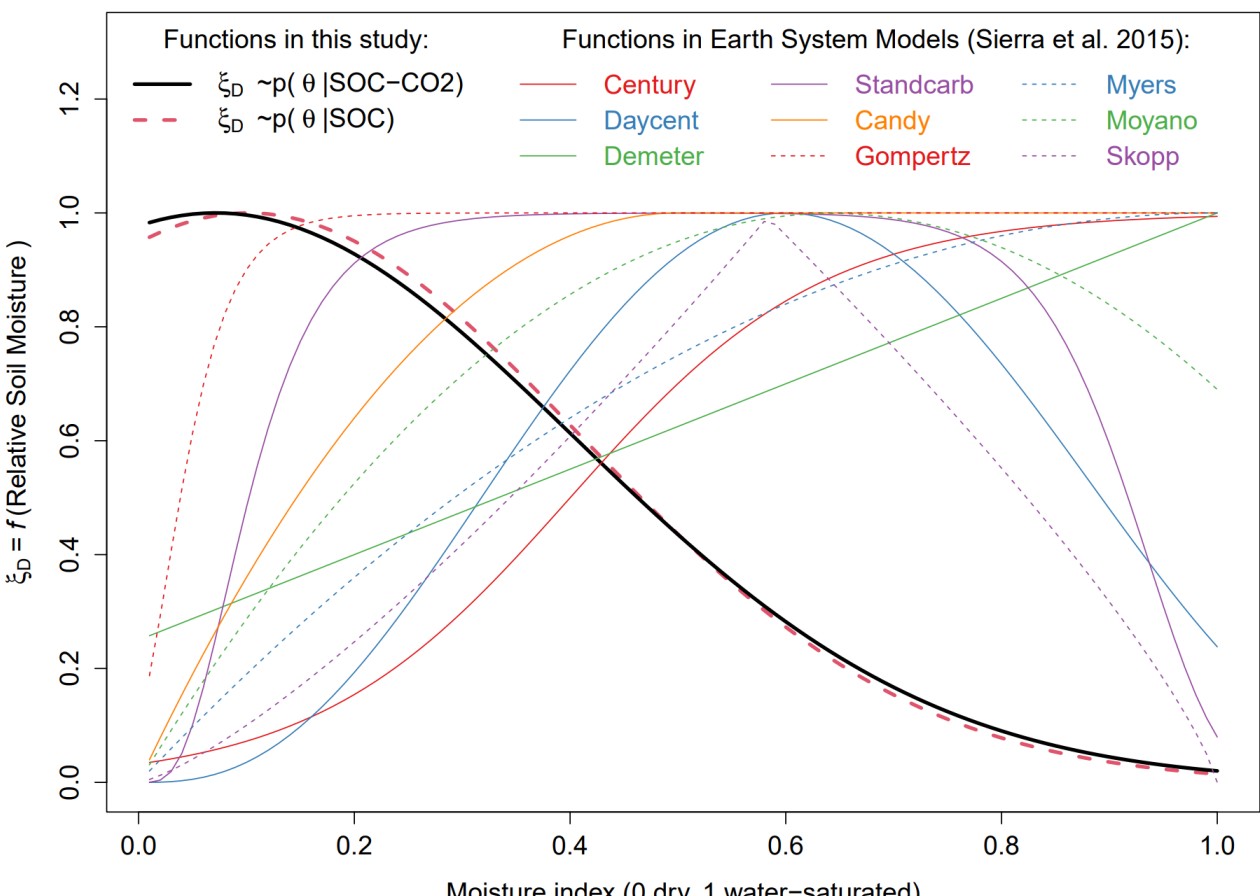



## 1    Introduction

Soil moisture and soil C stocks in boreal forests are higher in forested peatlands on frequently paludified organo-mineral soils

and peatlands on water-saturated organic soils than in well-drained forests on mineral soils (Weishampel et al., 2009; Ťupek et al., 2008; Bhatti et al., 2006; Hartshorn et al., 2003). Almost a quarter of the total terrestrial C (440 Pg C) stored in boreal moist and dry soils has accumulated since the last glaciation (Scharlemann et al., 2014) and is expected to create large C losses under warming climates (Hararuk et al., 2015). Moist organic soils are crucial for modelling dynamics of the global C cycle as they store five times more carbon than dry mineral soils (Leifeld and Menichetti 2018; Turetsky et al., 2015; Scharlemann

et al., 2014). However, soil organic carbon (SOC) stocks modelled by Earth System models (ESM) show large uncertainty due to structural model differences (Hashimoto et al., 2017; Hararuk et al., 2014, 2015; Todd-Brown et al., 2013), and differences in environmental drivers and their functional dependencies used by soil C models (Thum et al., 2020; Ťupek et al., 2019; Falloon et al., 2011).

Despite soil moisture being a dominant driver of variation in C dynamics (Humphrey et al., 2022), ESMs lack consensus on

the response of decomposition to soil moisture and temperature (Yan et al., 2018; Sierra et al., 2015; Falloon et al., 2011). The functional forms of the temperature and moisture modifiers of default decomposition rates among models disagree in their representation of extreme cold/dry and hot/wet conditions (Sierra et al., 2015). For example, the moisture decomposition dependency in the Yasso07 soil C model (Tuomi et al., 2011, 2009) is based on annual precipitation, has a functional form reaching saturation, and is uninformed about soil characteristics. The use of the saturation function is limited to well drained

soils as under wet or poorly drained forest soils such model results to underestimation of the C stocks (Dalsgaard et al., 2016, Ťupek et al., 2016). The soil module of the CENTURY model (Adair et al., 2008; Parton et al., 1996; Metherall et al., 1993) uses precipitation and basic soil data (bulk density, clay, and silt contents) to calculate soil moisture, which similarly to Yasso07 assumes saturation of decomposition rates. Other functional dependencies of moisture such as DAYCENT, Demeter, Standcarb, Candy, Gompertz, Mayers, Moyano, and Skopp assume all kinds of functional forms (e.g., Gaussian increase with

optimum and reduction of decomposition, continuous linear increase or with saturation, linear increase until optimum and linear reduction) (Kelly et al., 2000; Foley, 2011; Harmon and Domingo, 2001; Bauer et al., 2008; Janssens et al., 2003; Mayers et al., 1982; Moyano et al., 2013; and Skopp et al., 1990 as cited by Sierra et al., 2015). The wide variation in commonly used moisture functions may result from the variety of data from different soil types and climates used to constrain these moisture indices.

If environmental response functions were calibrated for mineral soils only, then these functions may not adequately represent responses in the moisture range characteristic of organic soils. For example, default response functions of soil C models cannot represent anoxic inhibition of decomposition rates in paludified peatland forest soils. However, the inhibition of decomposition can be accounted for by a reduction parameter such as "anerb" in CENTURY (Metherall et al., 1993). Due to variable water level and its determination of soil oxic/anoxic conditions and SOC accumulation in peatlands, peat SOC stocks are typically

modelled with models specifically developed for peatlands (Bona et al., 2020; Kleinen et al., 2012; St-Hilaire et al., 2010;



Frolking et al., 2010, 2001; Clymo 1992). However, for global applications on peatlands, the general soil models in ESMs can be modified for peat soil by adjusting parameters such as the hydraulic conductivity, as seen in models like JULES (Chadburn et al., 2022) and ORCHIDEE (Qiu et al., 2018), or by reducing decomposition rates for wetlands as in LPJ (Wania et al., 2010) and JSBACH (Goll et al., 2015). The land surface model JSBACH coupled with the Yasso soil C model adopts heuristic 65%

reduction of decomposition for wetlands (Kleinen et al., 2021; Goll et al., 2015). Using CENTURY model at the site-level, Raich et al. (2000) opted for improvement in modelled SOC of wetlands by modifying the environmental function for sites with insufficient drainage. This approach improved CENTURY compared to default Yasso07 in poorly drained forested peatlands in Sweden, though the SOC stocks of both models were still underestimated (on average by 10 and 13 kg C m$^{-2}$, respectively) (Ťupek et al., 2016). Similar magnitude of SOC underestimation of Yasso07 model with default dependency on

precipitation was also observed for poorly drained forest soils (e.g., gleysols and organic soils) in Norway (Dalsgaard et al., 2016).

We hypothesised that the SOC stocks and $CO_2$ emissions of mineral and organic (peat) soils can be modelled accurately by revising the original precipitation-based environmental modifier of a parsimonious model like Yasso07 with a function accounting for the reduction of decomposition based on the long-term near surface moisture. Near surface moisture is strongly

correlated with the ground water level depth in peatlands (Dimitrov et al., 2022) and the moisture values between mineral soil forests and peatlands are comparable on the same scale, which makes soil water content (SWC) a suitable variable for representing landscape moisture variation. Boreal forest SWC can either be measured in-situ or derived in high resolution using hydrological models (e.g., Leppä et al., 2020; Launiainen et al., 2019) and at larger scale by remote sensing and machine learning (Han et al., 2023). We aimed to develop the original Yasso07 model with global parameters as in Tuomi et al. (2011)

by adding a revised unimodal moisture-based environmental function. We then optimized this function using Bayesian data assimilation of measurements from a boreal forest-mire hillslope catena of mineral, organo-mineral, and organic soils, and tested whether we could correctly reconstruct observed SOC stocks and $CO_2$ emissions.

## 2 Methods

### 2.1 Study sites

Nine forest/mire site types in this study were situated along the hillslope from Vatiharju esker to Lakkasuo mire in southern Finland (61° 47', 24° 19') (Fig. 1) and formed a forest-mire ecotone, a gradient in soil moisture and nutrient status, vegetation composition, biomass production, and SOC stocks (Dimitrov et al., 2014). The sites were situated along a 450 m transect on a 3.3 % slope facing NE with a relative relief of 15 meters. The site typology described below was based on the vegetation composition reflecting site wetness, fertility, and location on the slope according to Finnish forest and mire classification

systems (Cajander 1949; Laine et al., 2004).



On the crest of the esker was a well-drained xeric Scots pine forest (CT – Calluna type) which changed down the slope to subxeric mixed Scots pine - Norway spruce forest (VT – Vitis-idaea type), and in mid-slope to mesic and herbrich Norway spruce dominated forest (MT – Myrtillus type, OMT - Oxalis-myrtillus type) together referred to as mineral soils upland forests. On the toe of the slope were forest-mire transitions on gleyic organo-mineral soils or mixed spruce pine birch forests

(OMT+ - Oxalis-myrtillus Paludified type, KgK – Myrtillus Spruce Forest Paludified type, and KR – Spruce Pine Swamp type). On the level were water-saturated sparsely forested mires on histosols (VSR1 and VSR2 - Tall Sedge Pine Fen types).

The understory or forest floor vegetation along the ecotone changed from being dominated by *Calluna* and *Vaccinium Vitis-idaea* dwarf-shrubs and typical forest mosses on the uppermost sites (CT, VT), to *Vaccinium myrtillus* dwarf-shrubs with herbs in the mid-slope (MT, OMT), *Vaccinium myrtillus* dwarf-shrubs with herbs and *Sphagnum* in the transitions (OMT+, KgK,

KR), and *Vaccinium oxycocus* and *Betula nana* dwarf-shrubs with *Menyanthes triofoliata, Carex* and *Sphagnum* species on the level (VSR1, VSR2) (Fig. 1). More detailed tree stand, soil and climate characteristics for these sites were reported by Ťupek et al. (2008, 2015).

## 2.2   Auxiliary measurements

Soil temperature, water content, and $CO_2$ emissions ($gCO_2$ m$^{-2}$ h$^{-1}$) were measured simultaneously during years 2004, 2005,

and 2006. The measurement campaigns were conducted in one or two days between 7 am and 6 pm weekly during the vegetative season of 2004 (July-November), 2005 (May-November), 2006 (May-September), and monthly during the non-vegetative season (December-April). The summer seasons of the years 2004, 2005, and 2006 showed exceptionally different monthly weather patterns. Data from Finnish meteorological station, located 3 km north-east from the ecotone in Juupajoki, showed that the summer season in 2004 was rainy and colder in comparison to long-term typically mild weather, in 2005

weather was typical, and in summer 2006 the weather was sunny and warm. The exceptional drought in 2006 caused by the lack of rain and increased temperatures in June and later July – early August (Gao et al., 2017) caused visible drying of the moss layer along all the sites of the ecotone. The 2006 summer drought ended with showers in mid-August and with more frequent rain in autumn the soil moisture recovered to a normal level.

### 2.2.1   Soil temperature and moisture conditions

The soil temperature was measured at depths of 5 cm ($T_5$, °C) with a portable thermometer, and the soil volumetric water content at depth of 10 cm ($SWC_{10}$, %, m$^3$ m$^{-3}$) in all sites with a portable ThetaProbe (Delta-T Devices Ltd) calibrated for each site type. The SWC calibration accounted for the bulk density/porosity of forest type specific soils (Ťupek et al., 2008, 2015). Because the forest-mire variation of soil organic layer bulk density was relatively small 0.34+/-0.07 g cm$^{-3}$ (porosity 74+/-5%) (Ťupek et al., 2015) the values SWC of top 10 cm were in the same order of magnitude between the forest/mire site types.

The instantaneous $T_5$ and $SWC_{10}$ measurements of the sites were interpolated and upscaled to monthly level based on regression models fitted between instantaneous forest/mire site measurements and continuous half-hourly data of $T_5$ and



SWC$_{10}$ from the nearby SMEARII station (study site of Helsinki University for measuring forest-atmosphere interactions located 6km NW from the forest-mire ecotone) (Hari and Kulmala, 2005).

### 2.2.2 Soil CO₂ emissions

Measurements of forest soil heterotrophic respiration (R$_h$, gCO$_2$ m$^{-2}$ h$^{-1}$, positive sign) were taken using opaque cylindrical chambers (30 cm diameter, 21.2 L) placed on metallic collars (30 cm in diameter) which were installed permanently into 30 cm soil depth. The collars' locations (12 for mineral soil forests, 9 for transitions, and 6 for mires, together 27) were selected to represent the spatial variation of each site type and the spatial variation along the forest-mire ecotone (e.g., dominant forest floor vegetation, microtopography, soil drainage, and nutrient status).

The aboveground forest floor vegetation inside each collar was removed at the time of collars installation and any plant regrowth of e.g., mosses was clipped approximately half-hour before the flux measurements. At the time of the collar installation the roots of the understory vegetation and trees were cut with a saw along the collars' diameter. The metallic collars installed to 30 cm soil depth prevented the regrowth of the roots. Due to the vast majority of tree and understory roots in boreal forest occurring in humus layer, the 30 cm depth was considered sufficient to cut the roots thus remove the signal of the root

autotrophic respiration from the net CO$_2$ emissions. In transitions and mires the depth of peat could be more than 30 cm (in range from 0.15 m in OMT+ to 1.2 m in VSR2 (Ťupek et al., 2008)) but the prevailing high-water levels (in range from 33 cm in OMT+ to 7 cm in VSR1 (Ťupek et al., 2008)) limit the root growth into the upper/sub-surface layer.

The soil CO$_2$ emissions were measured every 4.8 s during 80 s intervals with a portable infrared CO$_2$ analyser (EGM4, SRC-1 PP systems Inc.). We calculated CO$_2$ flux rates from the development of CO$_2$ concentration over time inside the chamber.

### 165 2.2.3 Soil organic carbon stocks

The soil data from the 2006 sampling up to 30 cm depth (Ťupek et al., 2015) were combined with additional soil sampling cores of up to 100 cm depth in October 2015 (3 per site). The bulk density, C and N concentrations for new samples were determined as in Ťupek et al. (2015). The SOC stock was a result of SOC content multiplied by a bulk density.

The SOC content (g cm$^{-3}$) of separate soil layers were interpolated for the whole profile with the fitted spline functions and
summed up for each depth and each forest/mire site (Fig. S1). The SOC content was similar in the upper-most humus layers of all forest/mire types (below 0.3 g cm$^{-3}$ in a layer 0-10 cm), but in the sub-surface level (10-30 cm) clearly doubled from uplands to transitions and mires (from below 0.2 to above 0.4 g cm$^{-3}$) (Fig. S1). In the soil layers below 30 cm the SOC content showed differences in degrees of magnitude (around 0, 0.01, and 0.1 g cm$^{-3}$ for forests, transitions, and mires, respectively) (Fig. S1).

### 175 2.2.4 Biomass of tree stand and understory vegetation

Brest height diameter and height of all Scots pine (*Pinus sylvestris*), Norway spruce (*Picea abies*), and Silver birch (*Betula pendula*) trees on each forest site type were measured in 2006. The biomass components for each species (leaves, branches,





stems, coarse-roots) were estimated with biomass conversion functions (Repola et al., 2008, 2009) and fine-roots with functions by Lehtonen et al. (2015). Forest floor plants from three 0.07 m$^2$ sample plots located nearby soil respiration

measuring collars were harvested for each forest/mire site type in June-July 2004 (Ťupek et al., 2008). Plants were separated to herbs, mosses, and shrubs, dried and weighed for each category and each sample plot. The stand density and the tree biomass increased from xeric (CT) and mesic upland forest sites (VT and MT) towards the herb-rich forest site (OMT) and transitions (OMT+, KgK, and KR), and decreased to very sparse canopy in peatlands/mires (VSR sites) (Fig. 1b). The understory aboveground biomass correlated negatively with the density of the canopy cover thus positively with the light intercepted onto

the forest floor (Ťupek et al., 2008).

## 2.3 Data analysis

### 2.3.1 Fitting Rh by nonlinear least-square regression models (NLS)

The Rh values were fitted separately for each forest/mire type to soil temperature and water contents. The $R_h$ NLS models were based on a $Q_{10}$ exponential function to $T_5$, soil temperature at 5 cm depth, adjusted with a response to $SWC_{10}$ which limits

$R_h$ outside the optimum soil water content (Davidson et al., 2012) (Eq. (1)).

$$R_{hij} = R_{href} 0.998^{\left(\text{SWC}_{\text{opt}} - \text{SWC}_{10}\right)^2} Q_{10}^{\frac{(T_5 - 10)}{10}} + \varepsilon_{ij} \qquad (1)$$

Where $i$ is forest/mire site type, $j$ is measured instantaneous soil respiration ($R_h$, g $CO_2$ m$^{-2}$ h$^{-1}$), $R_{h\text{ref}}$, $SWC_{opt}$, and $Q_{10}$ are parameters, and $\varepsilon_{ij}$ is the measurement error $j$ in $i^{th}$ forest/mire type. The parameter $SWC_{opt}$ represents the optimum soil moisture content for decomposition. The parameter values with their goodness of fit statistics are in Table S1. The shape of the curve

parameter (d in Davidson et al. (2012)) was set to 0.998 based on Ťupek et al. (2019). The $SWC_{opt}$ and $Q_{10}$ parameters of NLS model fitting (Table 1) were informed by the prior distribution of the same parameters (Table S1) derived through Bayesian data assimilation with the $SWC_{10}$ and $T_5$ functional dependency coupled with Yasso07 soil carbon model (described in detail below).

### 2.3.2 Yasso07 SOC and CO₂ modelling

Equilibrium SOC stocks of up to 1 m depth, SOC changes and soil $CO_2$ emissions (Rh) for the forest/mire types were modelled using the Yasso07 soil carbon model (Tuomi et al., 2009, 2011) with specific litter input and weather data in accordance with the method of Finnish greenhouse gas inventory (Statistics Finland, 2023). For the weather input, we first ran the Yasso07 model using the original formulation of the environmental function with precipitation and air temperature data, and then we ran the Yasso07 model fitted with the environmental modifier function based on soil temperature and moisture of the

forest/mire site types using the Bayesian data assimilation technique.

#### 2.3.2.1 Yasso07 soil C model

The Yasso07 is a semi-empirical process-based soil carbon model where soil C is divided based on the organic matter solubility into five pools ($C_A$, $C_W$, $C_E$, $C_N$, and $C_H$) from which three are fast (acid- (A), water- (W), and ethanol- (E) soluble), one is





slow (non-soluble (N)) and one is stable (humus (H)) (Tuomi et al., 2011). The rates of C decomposition in each pool and C

transfers between the pools are affected by climate. The model can be expressed mathematically as a set of differential

equations where decomposition of the entire structural matrix of C pools $C_A...C_H$ defined by default mass flow parameters

$\alpha_{A,W}...\alpha_H$ and decomposition coefficients $k_A...k_H$ ($A_{YS}$) is scaled by the time step dependent scalar of the environmental rate

modifier $\xi(t)$ Eq. (2).

$$\frac{dc(t)}{dt} = \begin{pmatrix} i_A \\ i_W \\ i_E \\ i_N \\ i_H \end{pmatrix} + \xi(t) \begin{pmatrix} -k_A & \alpha_{A,W}k_W & \alpha_{A,E}k_E & \alpha_{A,N}k_N & 0 \\ \alpha_{W,A}k_A & -k_W & \alpha_{W,E}k_E & \alpha_{W,N}k_N & 0 \\ \alpha_{E,A}k_A & \alpha_{E,W}k_W & -k_E & \alpha_{E,N}k_N & 0 \\ \alpha_{N,A}k_A & \alpha_{N,W}k_W & \alpha_{N,E}k_E & -k_N & 0 \\ \alpha_H k_A & \alpha_H k_W & \alpha_H k_E & \alpha_H k_N & -k_H \end{pmatrix} \begin{pmatrix} c_A \\ c_W \\ c_E \\ c_N \\ c_H \end{pmatrix} \qquad (2)$$

Where i defines a vector of initial carbon pools $i_A... i_H$, and subscripts in α indicate mass transfer pools (e.g., $\alpha_{A,W}$ defines mass

transfer from pool W to pool A). The total soil respiration or $CO_2$ efflux (Rh) can be expressed as a sum of vector of $C_A...C_H$

pools multiplied by a 5x5 diagonal matrix with a diagonal representing specific fractions of each C pool's decomposition rates

which are not transferred between the pools (Sierra et al. 2012).

The model was originally calibrated for running on annual time steps (Tuomi et al., 2009), but it can run on monthly steps with

monthly decomposition rates (1/12 of annual $k_A...k_H$), and monthly litter and climate data (Ťupek et al., 2019). Then $\xi(t_m)$ is

defined by a combined function of monthly air temperature ($T_m$) and 1/12 of annual precipitation ($P_a/12$) (Eq. (3)).

$$\xi_T(t_m) = e^{(\beta_1 T_m + \beta_2 T_m^2)} \left(1 - e^{\gamma \frac{P_a}{12}}\right) \qquad (3)$$

Where $\beta_1$, $\beta_2$, and $\gamma$ are parameters of the environmental function and $t_m$ is the monthly time step. To test our hypothesis of

running the model for a catena of soils with gradually increasing moisture content (from xeric to mesic, paludified, and

saturated), we re-defined the $\xi(t_m)$ function for the use with soil temperature and moisture data using Davidson et al. (2012)

empirical function ($\xi_D$, Eq. (4)).

$$\xi_D(t_m) = d^{(SWC_{opt} - SWC_{10})^2} Q_{10}^{\left(\frac{T_5 - 10}{10}\right)} \qquad (4)$$

Where $d$, and $SWC_{opt}$ parameters of the environmental function indicate the steepness and optimum of the hump-shaped

moisture function and the $Q_{10}$ parameter represents the exponential increase of the temperature function over 10 °C difference.

The Yasso07 model versions in this study run accordingly:

1.  the Yasso07.$\xi_{TW}$ version is the Yasso07 coupled with original $\xi_T$ (Eq. (3)) and with the original global parameter set
    (Tuomi et al., 2011) but with two k-rates parameter sets, (i) the original $k_A...k_H$ rates for application on mineral soils
    applied for mineral and organo-mineral soil forests (CT, VT, MT, OMT, OMT+, KgK, KR) and (ii) with an inhibitor
    reducing k-rates by 35% ($0.65k_A...0.65k_H$) for application on wetlands (Goll et al., 2015, Kleinen et al., 2021) applied for

mire sites (VSR1 and VSR2);




2. the Yasso07.$\xi_W$ wetland version is the same as the Yasso07.$\xi_{TW}$ but with a fine-tuned k-rates inhibitor to fit the SOC of mire sites (VSR1 and VSR2);

3. the Yasso07.$\xi_D$ version for soil moisture gradient from mineral to peat soils is the Yasso07 model coupled with $\xi_D$ (Eq. (4)) with the original global parameter set of the structural matrix and optimized parameters of $\xi_D$.

The initial equilibrium SOC stock ($C_o$) for each forest/mire type for the pre-trenching period was simulated analytically (Sierra et al., 2018) (Eq. (5)).

$$C_o = -\xi A_{YS}^{-1}\bar{u} \tag{5}$$

Where $\xi$ is the environmental modifier, $A_{YS}$ is a structural matrix formulation of Yasso07 model's differential equations, and $\bar{u}$ is the litter input (mean annual litter of foliage, branches, stem, stump, roots, and understory).

The Yasso07 model source code, used here, was built in R software (R Core Team 2021) on the platform of the SoilR package (Sierra et al., 2012) according to the mathematical description and parameters of Tuomi et al. (2011). The model outputs are monthly SOC stocks and soil $CO_2$ emissions. The model was run with data inputs of above- and below-ground litterfall (accounting for its chemical composition) and climate data (described in more detail below). Monthly outputs of heterotrophic soil respiration were downscaled to hourly values for comparison to Rh measurements.

**2.3.2.2 Climate and litter C input data for Yasso07 model**

The Yasso07.$\xi_{TW}$ was run with monthly air temperature and precipitation from the nearby Juupajoki weather station of the Finnish meteorological institute. The Yasso07.$\xi_D$ was run using site type specific continuous monthly $T_5$ and $SWC_{10}$ time series.

The litter C input of the forest/mire types (Fig. S2 and Fig. S3) used by Yasso07 was estimated as in Lehtonen et al. (2016)
based on turnover rates of tree stand biomass components (including fine- and coarse-roots, stump, branches, and foliage) and understory vegetation. The litter C input was separated into Yasso07 A, W, E, N pools according to the component and species (or species groups) specific A, W, E, N ratios taken from the literature (Berg et al., 1991a, 1991b, 1993; Gholz et al., 2000; Trofymow et al., 1998; Vávřová et al., 2009; Straková et al., 2010). The annual litter was distributed to monthly resolution by accounting for seasonal trends of foliage, fine-roots, and understory (Ťupek et al., 2019; Zhiyanski 2014) or evenly. The litter
input before trenching was assumed to represent the long-term average of the equilibrium state forest (Fig. S2a, Fig. S3). During trenching the severed fine- and coarse-roots made up the major component of the total litter (Fig. S2b) and resulted in a clear peak in the monthly litterfall time series (Fig. S3). After trenching the monthly litter levels decreased as the sum of components excluded the roots (Fig. S2c, Fig. S3).

**2.3.2.3 Bayesian SOC and $CO_2$ data assimilation**

The Bayesian posterior uncertainty provides updated information on parameter values based on pre-existing information on the parameters and the data through the likelihood function (Speich et al., 2021). The d, $Q_{10}$, and $SWC_{opt}$ parameters of the $\xi_D$



(Eq. (4)) coupled with Yasso07 model were optimized on the level of the forest-mire ecotone using Bayesian data assimilation technique (Luo et al., 2011; Hartig et al., 2012; Speich et al., 2021) with observed SOC stocks and monthly Rh data of forest/mire types with prior information on best parameter values obtained from a purely empirical NLS model (Table 1) and

the defined parameter range in Table S1. During the optimization, the Yasso07.$\xi_D$ model was first run only with observed SOC stocks and second with both SOC stocks and Rh data combined obtaining a probability distribution of model parameters of $\xi_D$ (the posterior uncertainty $p(\theta|y)$ conditional on the observations (y) and prior knowledge on the parameter values $p(\theta)$). The sum of the probability density for the target parameter set ($\theta$) between the model predictions and observations was maximized for the best agreement using the likelihood defined by a modified Laplace probability density function $p(y|\theta)$ (the probability

of observing the data y with the model parameters set $\theta$) where we allowed the width of the distribution to be affected by the observed SOC and Rh values (Eq. (6)).

$$p(y \mid \theta) = \prod_{j=1}^{2} \prod_{i=1}^{N_j} \frac{1}{2(a_j + b_j x_{j,i})} \; e^{\frac{-|(x_{j,i} - \mu_{j,i})|}{a_j + b_j x_{j,i}}} \tag{6}$$

where $\mu_{j,i}$ is the observed $j^{th}$ variable (e.g., SOC or SOC and $CO_2$) of $i^{th}$ observations, $x_i$ is the modelled prediction, n is the total number of observations, and a, b are parameters affecting the width of the distribution. In the combined SOC and $CO_2$

likelihood, the likelihood function $p(y \mid \theta)$ was then the multiplication of the distributions of SOC and $CO_2$ at all observation times. To account for the impact of different timescales and the number of observations on the residual distribution of the SOC and $CO_2$ data sources (Xu et al., 2006), the $CO_2$ distribution was weighted by the empirically sought value (0.1).

The model parameters of $\xi_D$ and $p(y|\theta)$ were sampled from an assumed uniform distribution within their prior ranges (Table S1). Posterior probability distributions of parameters (Table 2) were derived by using the differential evolution (DEzs) Markov

Chain Monte Carlo (MCMC) sampler (ter Braak and Vrugt, 2008) with the runMCMC function from the BayesianTools package in R (Hartig et al., 2012) and by computing three chains in parallel. The convergence of MCMC runs was evaluated using Gelman–Rubin multivariate potential scale reduction factor (psrf) (Brooks and Gelman, 1998). The MCMC simulation was considered converged if psrf was below 1.05 for all parameters (1.028 and 1.048 for $p(\theta|SOC)$ and $p(\theta|SOC\text{-}CO2)$, respectively). Trace plots of MCMC runs for target parameters showed effective sampling and unimodal parameter density

with clearly defined peaks. The differences in parameter uncertainties (difference between 97.5% and 2.5% quantiles of the 95% confidence interval) were not significant (p = 0.99) when evaluated with a Welch Two Sample t-test between two posterior distributions $p(\theta|SOC)$ and $p(\theta|SOC\text{-}CO2)$ (Table 2).

### 2.3.2.4 Performance evaluation of Yasso07.$\xi_{TW}$ and Yasso07.$\xi_D$

The performance of Yasso07 model versions (i) Yasso07.$\xi_{TW}$ and (ii) Yasso07.$\xi_D$ with $\xi_D$ parameter set $\theta$ from two posterior

distributions, $p(\theta|SOC)$ and $p(\theta|SOC\text{-}CO_2)$, was evaluated with the modelled SOC and $CO_2$ outputs against the observed data in the forest mire-ecotone with the coefficient of determination ($R^2$), the mean absolute error (MAE), mean bias error (MBE), the root-mean-square error (RMSE), the Akaike information criterion (AIC) for considering the number of model parameters





in the error calculation as in Abramoff et al. (2022), and the fitted linear trends of normalized SOC and $CO_2$ model residuals with observations against $T_5$ and $SWC_{10}$ data.

# 3    RESULTS

### 3.1    Distributions of SOC stocks and Rh in relation to SWC

The SOC stock measurements (to a depth of 1 m) in forest-mire ecotone were distributed in range between 20 in well-drained soils of upland forests and 125 kg C m$^{-2}$ in poorly drained soils in peatlands/mires (Fig. 2). The SOC stock values strongly correlated with the long-term moisture levels. The median Rh values ranged between 0.4 and 0.6 g$CO_2$ m$^{-2}$ h$^{-1}$ for upland forests, 0.4 and 0.5 g$CO_2$ m$^{-2}$ h$^{-1}$ for forest-mire transitions, and 0.3 and 0.4 g$CO_2$ m$^{-2}$ h$^{-1}$for mires (Fig. 2). The forest/mire site type differences in median Rh values expressed per m$^2$ were small and poorly correlated with the mean soil moisture levels. The Rh expressed in ppm as the emitted C fraction of total SOC (C/SOC) was highly correlated with moisture and values per unit peat in mires were much lower than in upland forests (Fig. 2).

### 3.2    Distribution of Rh in climate space of soil T and SWC

The site-specific time series of hourly $R_h$ measured in the forest/mire ecotone during years 2004, 2005, and 2006 followed a typical seasonal pattern of temperature and was distributed in range between 0.08 and 1.6 g$CO_2$ m$^{-2}$ h$^{-1}$ depending on the corresponding soil temperature and moisture conditions (Fig. 3). The Rh values were generally larger during wet years than during a typical year, and lowest during dry years (Fig. 3).

The $T_5$ and $SWC_{10}$ values showed a typical seasonal variation (in range between around 0 and 20°C, driest in summer and wettest in late autumn/spring) (Fig. 2b and 3c). The $T_5$ showed similar magnitude among the forest/mire sites, whereas the $SWC_{10}$ increased from driest (upland forest) to intermediate (forest-mire transition), and from upland to lowland for the wettest (mire) sites located on the slope (Fig. 3). The SWC values of the top 10 cm were comparable in the same order of magnitude between the forest/mire site types because the forest-mire variation of the soil organic layer bulk density was relatively small 0.34+/-0.07 g cm$^{-3}$ (porosity 74+/-5%). The forest-mire ecotone soil moisture at 10 cm depth ranged from 5% to 91%. The minimum, maximum and optimum SWC at 10 cm depth between forests, transitions, and mires clearly differed showing the gradient of increasing moisture from forests to mires (Fig. 2a, Fig. 3). Due to highly variable weather (wet, typical, and dry year) all ecosystems experienced periods of extremely low and high $SWC_{10}$ values. The $SWC_{10}$ of upland forest ranged between 5 and 25%, between 17 and 70% in transitions and mires between 49 and 91% (Fig. 3). The variation of soil temperature at 5 cm depth along the ecotone was similar between the forest/mire types and ranged between -3 and 22 °C (Fig. 3). The $R_h$ values during dry 2006 years were in comparison to previous years clearly reduced mostly in upland forest and forest-mire transitions while in mires the soil respiration was comparable to previous seasons (Fig. 3). The effect of spatially prevailing SWC levels of forest/mire types was not clearly reflected in Rh values when expressed in g$CO_2$ m$^{-2}$ h$^{-1}$ (Fig. 2)




(unless expressed as a C fraction of SOC) and short-term SWC variation impacted the typical seasonal levels of Rh values mainly during the extreme events (rainy summer period in wet years, or drought summer period in dry years) (Fig. 3).

### 3.3    $\xi_D$ optimized with Yasso07.$\xi_D$

The optimization of $\xi_D$ (Eq. (4)) coupled with Yasso07 showed that in the catena of mineral and organic soils of the boreal forest-mire the optimum moisture content for decomposition and $CO_2$ emissions was in well drained mineral soil forests ($SWC_{opt}$ medians between 5.5 and 9.5 %, Table 2, Fig. 4). The increase in long-term soil moisture content above 10 % inhibited carbon mineralization more than dryness did (Fig. 3b). The decomposition rate outside the moisture optimum reduced decomposition similarly for the two data sources (SOC and SOC-$CO_2$) used for calibration. The $\xi_D$ $SWC_{opt}$ parameters derived with Bayesian data assimilation using SOC or SOC-$CO_2$ dataset for the forest-mire ecotone were comparable to $SWC_{opt}$ parameter of the NLS model based only on soil heterotrophic respiration data. Both optima were found in relatively dry conditions of the forest-mire ecotone in well drained mineral soils (Tables 1 and 2).

The optimization using two data sets showed that the temperature sensitivity $Q_{10}$ parameter varied between 3.5 based on SOC-$CO2$ data and 4.3 when based on only SOC data (Table 2, Fig. 4a). The two $Q_{10}$ functions showed a similar increase with $T_5$ until 10 °C, above which the $Q_{10}$ function calibrated with SOC data increased 15% more than the function based on SOC-$CO2$ data (Fig. 4a). Similar patterns were observed for $\xi_D$ in the climate space of $T_5$ and $SWC_{10}$ where above 10 °C the values of $\xi_D$ increased for $p(\theta \mid SOC)$ compared with $p(\theta \mid SOC-CO_2)$ (Fig. 4c and 4d). The stronger temperature sensitivity of $\xi_{D,p(\theta \mid SOC)}$ to $\xi_{D,p(\theta \mid SOC-CO2)}$ resulted in a more pronounced increase in decomposition rates. $\xi_D$ was greater than 1 in drier conditions with a wider and higher increase of $\xi_{D,p(\theta \mid SOC)}$ than $\xi_{D,p(\theta \mid SOC-CO2)}$ above 10 °C. The SOC and SOC-$CO_2$ based $\xi_D$ $Q_{10}$ values for the forest – mire ecotone (medians 4.0 or 3.6, respectively, Table 2) showed higher sensitivity to temperature increase than the $Q_{10}$ parameter obtained from NLS model (2.6, Table S1). The NLS-based $Q_{10}$ values fitted for forest type groups separately showed increase from mires on organic soils (1.77), to organo-mineral soils in forested peatlands (1.98), to forests on mineral soils (2.39) (Table 1).

### 3.4    Performance of Yasso07.$\xi_{TW}$, Yasso07.$\xi_W$ and Yasso07.$\xi_D$

The model performance evaluation showed that the soil water and temperature modifier $\xi_D$ coupled with Yasso07 model (Yasso07.$\xi_D$) outperformed the original Yasso07 environmental function even after 65 % reduction of decomposition rates for wetlands was applied (Yasso07.$\xi_{TW}$) (Table 3, Fig. 5). Although, the Yasso07.$\xi_{TW}$ model version accurately predicted SOC stocks of mineral soil forests (CT…OMT), it heavily underestimated the SOC stocks of organo-mineral forested peatlands and mires (OMT+…VSR2), thus it showed the most biased model performance metrics (highest RMSE, MBE, MAE, AIC and lowest $R^2_{adj}$) among the model versions compared (Table 3). Reduction of decomposition rates of 65% for mires in Yasso07.$\xi_{TW}$ was not sufficient to simulate their SOC stocks as simulated SOC of mires were only about 10 % of measured values (Figs 5a, 5b, and 5c). The SOC simulations for VSR mires with Yasso07.$\xi_W$ would have required as much as a 96%





reduction of the decomposition rates. The optimized Yasso07.$\xi_D$ model version accurately simulated SOC stocks throughout
360   the forest – mire ecotone.

The version Yasso07.$\xi_{D, p (\theta \mid SOC)}$ outperformed Yasso07.$\xi_{D, p (\theta \mid SOC-CO2)}$ when evaluated against SOC data and vice versa when
the models were also evaluated against $CO_2$ data (Table 3, Fig. 5). The normalized SOC residuals of two Yasso07.$\xi_D$ models
did not show any $T_5$ or $SWC_{10}$ trends (Figs 5a, 5b, and 5c).

The soil $CO_2$ emissions simulated with original Yasso07.$\xi_{TW}$ agreed unexpectedly well with observed Rh values (Table 3,
365   Figs. 5d, 5e and 5f) outperforming the Yasso07.$\xi_{D, p (\theta \mid SOC)}$ version which overestimated the Rh for high temperatures due to
higher $Q_{10}$ (Table 2, Fig. 4a, Fig 5e). On the other hand, the Rh simulated with Yasso07.$\xi_{D, p (\theta \mid SOC)}$ performed similarly as
Yasso07.$\xi_{TW}$ in terms of RMSE (same RMSE 0.12 g $CO_2$ $m^{-2}$ $hour^{-1}$ for both models). Although, the $R^2_{adj}$ of Yasso07.$\xi_{D, p (\theta \mid}$
$_{SOC)}$ was better compared with Yasso07.$\xi_{TW}$ (0.39 and 0.15, respectively), and the AIC of Yasso07.$\xi_{D, p (\theta \mid SOC)}$ was worse (-
3095 and -4138, respectively) (Table 3).  The normalized modelled residuals by absolute values of Rh measurements showed
that both Yasso07 model versions (Yasso07.$\xi_{TW}$ and Yasso07.$\xi_{D, p (\theta \mid SOC)}$) showed small biases of basal Rh in extreme, very
cold and in very warm temperatures (Fig. 5e). The normalized $CO_2$ residuals of Yasso07.$\xi_{D, p (\theta \mid SOC-CO2)}$ plotted versus
temperature did not show any bias, and the normalized $CO_2$ residuals evaluated against with $SWC_{10}$ did not show any bias for
any of the models (Fig. 5f).

## 4   Discussion

The Yasso07 model (Tuomi et al., 2011) coupled with a revised and optimized soil temperature and moisture empirical function
$\xi_D$ (Eq. (4), Fig. 4), successfully reconstructed observed variation of SOC stocks and soil heterotrophic $CO_2$ emissions with
increasing soil wetness in mineral, organo-mineral, and peat soils in boreal forest (Fig. 5). The Bayesian MCMC data
assimilation was effective in finding landscape moisture optima and has been successfully used in land surface soil C modelling
(e.g., Xu et al., 2006; Hararuk et al., 2014).

Our application of Yasso07 models on the hillslope accounted for the continuity in moisture conditions which was reflected
in the modelled gradient of mineral and peat soils carbon stocks. The Yasso07 model initially developed for mineral soils was
improved for application in peatlands by accounting for the soil temperature and volumetric moisture, as these are better
predictors of heterotrophic respiration than air temperature and precipitation (Jian et al., 2022). Although, the empirical
function $\xi_D$ used here was heuristic, it implicitly accounted for prevailing intrinsic micro-scale processes on the hillslope
controlling Rh and SOC accumulation e.g., plant and microbial communities, long-term and short-term limitation of oxygen
and substrate with moisture (Davidson et al., 2012).

The $\xi_D$ being able to simulate gradually increasing SOC stocks from mineral to organic soils makes it a preferable rate modifier
for the Yasso07 model, instead of simply adjusting decomposition with a reduction constant for wetlands (e.g., Goll et al.,
2015; Kleinen et al., 2021), which underestimated the SOC stocks of peatlands (Yasso07.$\xi_{TW}$ in Fig. 5). In this study, the





constant 96% reduction (0.04*k-rates) was proposed for the existing Yasso07.$\xi_{TW}$ for more accurate SOC modelling in mires, a value comparable to rates of anaerobic decomposition (Schuur et al., 2015). Currently, 80% reduction of rates is used for water-saturation in an updated moisture modifier in the JULES model (Chadburn et al., 2022). Although JULES reduces decomposition linearly from the maximum rate 1 at the moisture optimum (30% - 75% SWC) to a reduced rate 0.2 in water-saturated peat soils.

## 4.1     The moisture response

The use of gradually changing near surface soil moisture avoids biases in land surface modelling related to ignoring high SOC stocks of organo-mineral soils of forested peatlands (Dalsgaard et al., 2016, Ťupek et al., 2016), e.g., forest-mire transitions (Fig. 1, and Fig. 2). Obviously, modelling decomposition rates accurately with moisture functions accounting for microbial processes requires correctly defined shape parameters of the response curve (steepness of increase/decrease, optimum, and its

range) (Yan et al., 2018; Moyano et al., 2012, 2013; Manzoni et al., 2012). However, uncertainty in functional moisture - soil respiration dependencies are high (Sierra et al., 2015; Falloon et al., 2011) and dependencies vary with the soil properties, e.g., SWC optimum increases for soils with higher organic C content (from 30% to 75% SWC, Moyano et al., 2012, 2013) as also observed in this study (SWC$_{opt}$ increased from 18 and 67% between forests and mires, Table 1). The nonlinear least-square regressions of soil respiration fluxes for the whole forest-mire ecotone showed the SWC$_{opt}$ at 31% (Table 1) which agreed with

the optimum found in datasets of sites from a larger moisture range (found in 50% water-filled pore space (WFPS) and corresponding to around 31% SWC assuming mean porosity of 62%, Hashimoto et al., 2011).

The impact on decomposition of the NLS function and the $\xi_D$ function incorporated into Yasso07 soil C model was comparable (e.g., both found the moisture optimum in dry soils of forest-mire ecotone). Although, the NLS model for the soil temperature and moisture function showed a relatively small difference in Q$_{10}$ and SWC$_{opt}$ parameters (2.6 and 31, respectively) compared

to these parameters in $\xi_D$ (3.4-4.0 and 5.5-9.4, respectively) (Table 1 and 2). Different Q$_{10}$ and SWC$_{opt}$ parameter values between fitted functions of NLS and Yasso07 could be expected due to a different basal respiration in the two modelling approaches; in NLS, basal respiration is a constant parameter (Rh$_{ref}$ Eq. (1), Table 1) but in Yasso07 it was a dynamically time-dependent variable, reflecting Yasso07 default decomposition rates, and the monthly change in SOC and litter input (Eq. (2)). In terms of the Yasso07 model constants, if temperature and moisture conditions are favourable for organic matter stabilization

then the $\xi_D$ is reduced (Fig. 4) which reduces decomposition rates of fast and slow C pools, reduces their $CO_2$ emissions, and increases C storage. The forest-mire sites' heterotrophic respiration per unit of area did not show a clear difference between well-drained and water-saturated soils whereas the C mineralization per unit SOC was clearly reduced in soils with high legacy field soil moisture (Fig. 2). Reduction in decomposition rates in the environmental gradient from low to higher field moisture, indicates possible a difference in the soil C stabilization mechanisms under low- and high-water content (Das et al., 2019).

The $\xi_D$ function and its reduction with increasing wetness from dry soils was based on a large range of forest/mire soil C stocks (between 11 and 134 kg C m$^{-2}$) reflecting a spatial long-term moisture gradient between forests and mires (Fig. 2) and its short-



term moisture and $CO_2$ dynamics over years with contrasting climate (Fig. 3). The soil respiration data from three years covered exceptionally contrasting wet and dry summers and likely captured a full range dependency on the soil moisture induced by short-term weather variation in a spatial/long-term forest-mire gradient in soil moisture, soil C pools, vegetation, and microbial

composition. The short-term deviations in respiration indicative of wetting/drying cycles (Barnard et al., 2020; Patel et al., 2021) could be seen by the respiration increases in wet summers or during and after a period of drought (Figure 3). Thus, the $\xi_{D,p(\theta|SOC-CO2)}$ curve calibrated with highly variable SOC and $CO_2$ data from a forest-mire ecotone represented a mean robust moisture-decomposition dependency smoothing short-term weather dependent fluctuations with the spatial variation of organic matter decomposition across ecological gradients. This function could meet the land surface modelling criteria for spatial

accuracy on small scales but also cost efficiency for running or forecasting the C dynamics at large scales (Luo and Schuur, 2020).

The $\xi_D$ function's $SWC_{opt}$ found in dry conditions and reduction of default decomposition rates (k) with increasing soil wetness contrasted with responses from the short-term laboratory incubation soil respiration studies (weeks, months) showing increase in decomposition from dry conditions until reduction in very wet (Sierra et al., 2017; Moyano et al., 2012, 2013; Kelly et al.,

2000; Skopp et al., 1990; Yan et al., 2018). The $\xi_D$ optimized with SOC and $CO_2$ data ($\xi_{D,p(\theta|SOC-CO2)}$) showed that the optimum/maximum decomposition rate in the forest-mire ecotone was in dry conditions below 10 % of mean long-term near surface SWC (around 16 % WFPS, corresponding to xeric ad sub-xeric forest site types) ($SWC_{opt}$ parameter in Table 2, Fig. 4b) whereas the moisture optimum of studies based only on soil respiration was around 40% - 60% (Fairbairn et al., 2023; Moyano et al., 2013; Kelly et al., 2000; Skopp et al., 1990; Yan et al., 2018).

The major reason for a lower $SWC_{opt}$ in $\xi_D$ in comparison to other studies was due to the combination of field measurements (mineral soil and peat SOC stocks, and soil heterotrophic $CO_2$ respiration, litter input) and extreme weather variability of three years coupled into soil C model optimization in contrast to abovementioned dependencies limited to relatively short-term responses of only soil respiration from mainly mineral soils and that were incubated in laboratory conditions. In optimizing model performance with a multi-variable data set, Keenan et al. (2013) found that a combination of data with fast and slow

turnover (e.g., soil respiration and soil carbon stocks) leads to the largest improvement in model performance. The Yasso07.$\xi_D$ based only on slow (SOC) was as good as constraining with SOC and $CO_2$, as both approaches accurately observed soil $CO_2$ emissions and SOC stocks along the site types of the forest-mire ecotone with no clear bias in residuals (Fig. 5). Thus, in a catena of mineral and peat soils of forest-mire ecotone, and in the combined measured SOC and $CO_2$ data assimilation in $\xi_D$ (9 and 2369, respectively), the relatively small number of SOC stocks (9 forest/mire types) largely determined the SWC

response form reflecting both a spatial moisture gradient and its temporal variation.

The $SWC_{opt}$ discrepancy of the $\xi_D$ function highlights the difference between (1) the responses from the field-based or long-term soil respiration measurements reflecting moisture responses of older, stabilized and slowly decomposing SOC, and (2) the short-term incubation-based soil respiration studies which predominantly capture decomposition of newly available, labile and rapidly decomposing, SOC pool (González-Domínguez et al., 2022; Huang and Hall, 2017). We observed an immediate





high $Q_{10}$ sensitivity of microbial respiration to temperature in wetter soil. However, the reverse can be observed over longer periods of incubation (i.e., high $Q_{10}$ in drier soil) (Zhou et al., 2019). In our study, this long-term effect is indicated by $Q_{10}$ increase from lower to higher values from wet mire to drier forest sites (Table 1). Despite the difference in SOC pool response to soil moisture in relation to different timescales of field and laboratory data, the moisture optimum of the unimodal the $\xi_D$ function found in well-drained mineral soils well captured high soil $CO_2$ emissions, such as during the rainy summer in xeric

and sub-xeric forest sites and during the rewetting period after drought (Fig. 3) without apparent bias in the Rh predictions (Fig. 5). The enhanced C mineralization can occur during periods of elevated moisture under Fe reduction when microbes can access previously protected labile C (Huang and Hall, 2017).

Although, the moisture representation of the $\xi_D$ environmental function was accurate at the forest-mire ecotone level, at the forest site level the contrasting respiration responses to moisture (i.e., either respiration reduction during soil drying or

increased $CO_2$ emissions with rewetting (Barnard et al., 2020; Patel et al., 2021) for dry soils or the opposite for wet soils), were likely not captured sufficiently by the curve which was symmetrical around the optimum. The relatively small reduction of respiration in relation to dryness could thus be partly driven by the form of the bell-shaped function and by the prevailing soil respiration from the deeper soil layers even when the near surface moisture was extremely dry. Soil C modelling might be further improved using a moisture response that accounts separately for microbial growth respiration with increased water

availability, and for oxygen limitation in soil reaching water saturation (Sierra et al., 2015). However, as the aim of the moisture response used in this study was applying the above concepts in a cost-efficient way using an empirical function with only two easily interpretable parameters (shape and optimum) (Davidson et al., 2012), the mathematical representation of the moisture function with increased complexity still needs to be evaluated in further studies testing different functional forms with larger regional data.

## 475 4.2    The temperature response

The original air temperature-based modifier in Yasso07 was replaced by the Arrhenius type temperature function driven by soil temperature. This function was found to best represent the enzyme kinetics under unconstrained substrate and oxygen (Sierra et al., 2017). The optimized temperature function with combined SOC and $CO_2$ data produced accurate modelled values (Fig. 5). The $Q_{10}$ values around 4.0 were comparable with the well-, moderately-, and poorly- drained forest soils for similar

climates (Chen et al., 2020; Davisdon et al., 1998; Karhu et al., 2010; Pumpanen et al., 2008).  However, the optimization of the Arrhenius type temperature response only with SOC data resulted in higher $Q_{10}$ (4.0) compared to SOC and $CO_2$ based $Q_{10}$ (3.6) (Table 1), which caused small overestimation of decomposition rates above 10 °C, and mismatch with summer respiration peaks (Fig. 5). Thus, due to the comparable predictive power of soil and air temperature (Jian et al., 2022), the original Gaussian temperature dependency could be more accurate than Arrhenius response for the optimization with only SOC data (Tuomi et

al., 2008).



Underestimated respiration during winter (Fig. 5) could be also caused by a scarcity of winter field $CO_2$ measurements potentially resulting in larger random errors (e.g., due to difficulties of measuring relatively small respiration fluxes during soil freezing/thawing cycles, measurements on soil covered by snow layer, and reduced precision of gas analysers during measurements in lower temperature range).

**5    Conclusions**

The Yasso07 soil carbon model was developed and parameterized at global scale for mineral soils; however, it has also been applied for land surface modelling coupled with the JSBACH model with a 65% reduction of default decomposition for wetlands. At the forest site level, we evaluated the performance of the Yasso07 model with an original climate modifier based on air temperature and precipitation against the model coupled with a revised environmental modifier based on soil temperature

and moisture. We found that the Yasso07 model coupled with revised climate dependencies performed similarly for mineral soils but outperformed the original configuration with the JSBACH modification for undrained peatland soils.

The optimization of moisture dependency conducted in this study accounted for both a spatial moisture gradient and its temporal variation. The moisture optimum at dry soils has not changed depending on whether the function was optimized using both slow (SOC) and fast ($CO_2$) turnover data or only slow (SOC) data.

The SOC stocks in peatland forests were an order of magnitude larger in comparison to forests on mineral soil. On a landscape level, these peatland SOC stocks had the largest influence on moisture optimum, when they were included along with fluxes in optimization. The function implicitly accounted for relative contribution of C fluxes from short term biogeochemical processes in a long-term SOC accumulation. Thus, for accurate estimates of the boreal forest soil carbon pools with Yasso07 model, the SOC accumulation related to inhibition of decomposition with increasing wetness was more pronounced than the

one related to dryness.

This study illustrated the limitation of the default moisture functions used for peatland forest soil C modelling. Also, the unimodal function with a proposed moisture optimum in well-drained mineral soils needs further evaluation with regional boreal forest data. If the dry soil moisture optimum of litter decomposition in boreal forest landscape proves to be robust, then in the future warmer and drier climates the boreal forest could be expected to enhance soil C emissions to the atmosphere due

to water level drawdown of presently water-saturated peat soils with large C stocks. However, rewetting of previously drained peatlands could be expected to reduce soil C emissions, turning SOC loss to long-term C sequestration.

**6    Data and code availability**

The input data (soil $CO_2$ fluxes, soil temperature and moisture, air temperature and precipitation, tree stand and understory inventory, and soil C stocks) as well as the analysis (R codes) needed to run the Yasso07 model versions and reproduce the

results of this study are available on Zenodo, https://doi.org/10.5281/zenodo.8111475.



## 7    Author contribution

BT designed the hypothesis, collected data (soil respiration, micrometeorology, tree and understory inventory, and soil C), and carried out the analysis (input data preparation, the model reformulation on SoilR platform, the model update, calibration, and evaluation). KM contributed to design of the ecological gradient of study sites, and design of measurements of soil respiration

and micrometeorology. AY contributed to soil sampling and soil C data preparation. AL contributed to codes on biomass and litter estimation. XT contributed to formulation of likelihood for model calibration. BT prepared the manuscript with contributions from all co-authors.

## 8    Competing interests

We have no conflict of interest to declare.

## 9    Acknowledgments


The study was partially supported by the SOMPA project (Novel soil management practices - key for sustainable bioeconomy and climate change mitigation) funded by the Strategic Research Council at the Academy of Finland (no. 312912 and 336570) and the HoliSoils project (Holistic management practices, modelling and monitoring for European forest soils) funded by European commission (EU Horizon 2020 Grant Agreement No. 101000289).

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





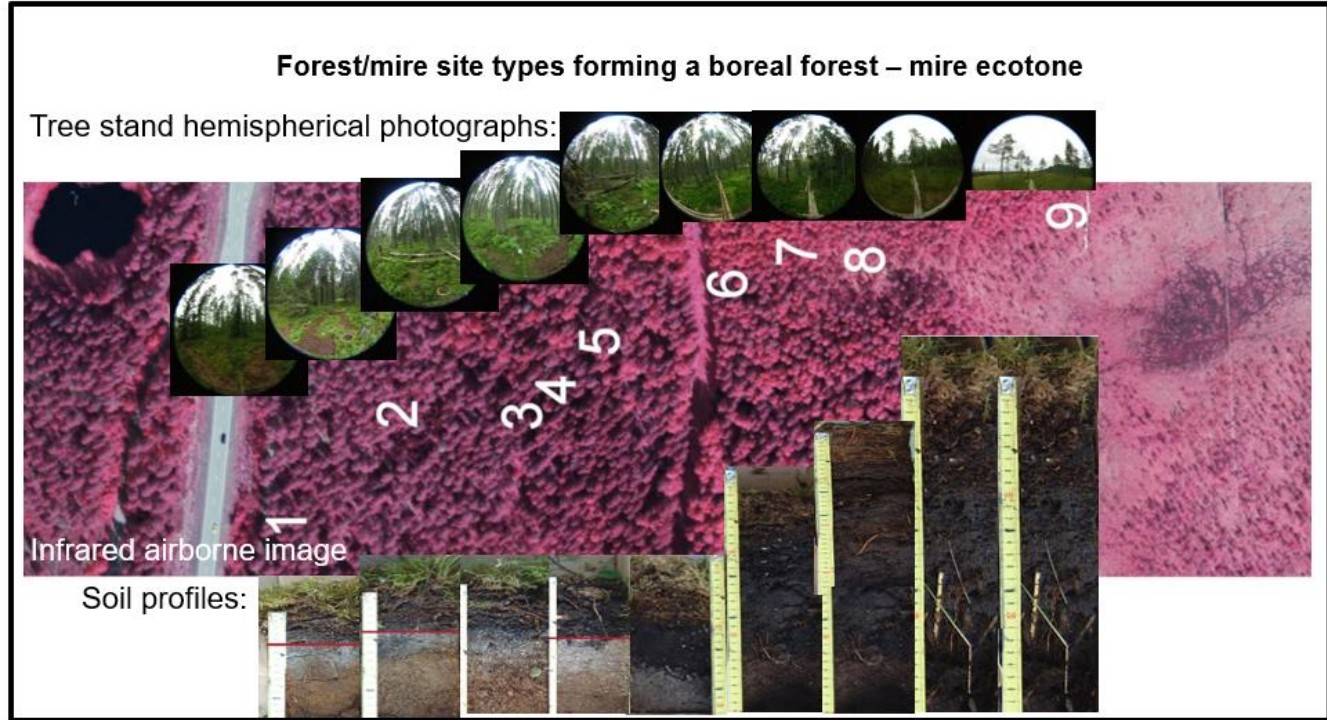

Figure 1. Infrared areal image showing the location of nine studied forest/mire types forming a transect of approximately 450 m on the northern hillslope in Finland (61° 47', 24° 19'). The series of hemispherical images of forest stands on the top of the aerial image show the increasing gradient in the canopy openness from upland forests (left) to mires (right). The series of soil profiles show the increasing gradient of the organic layer depth. The images in the series arranged from left to right mimic the site type location on the slope from the hill to depression. Sites range from upland (1) xeric, (2) sub-xeric, (3) mesic  and (4) herb-rich forest types (CT - Calluna, VT - Vitis Idaea, MT - Myrtilus, OMT - Oxalis-Myrtillus), through paludified forest - mire transitions (5 - 7) (OMT+ - Oxalis-Myrtillus Paludified, KgK – Myrtillus Spruce Forest Paludified, KR – Spruce Pine Swamp), to sparsely forested mires/peatlands in depression (8 - 9)  (VSR1 and VSR2 - Tall Sedge Pine Fen).



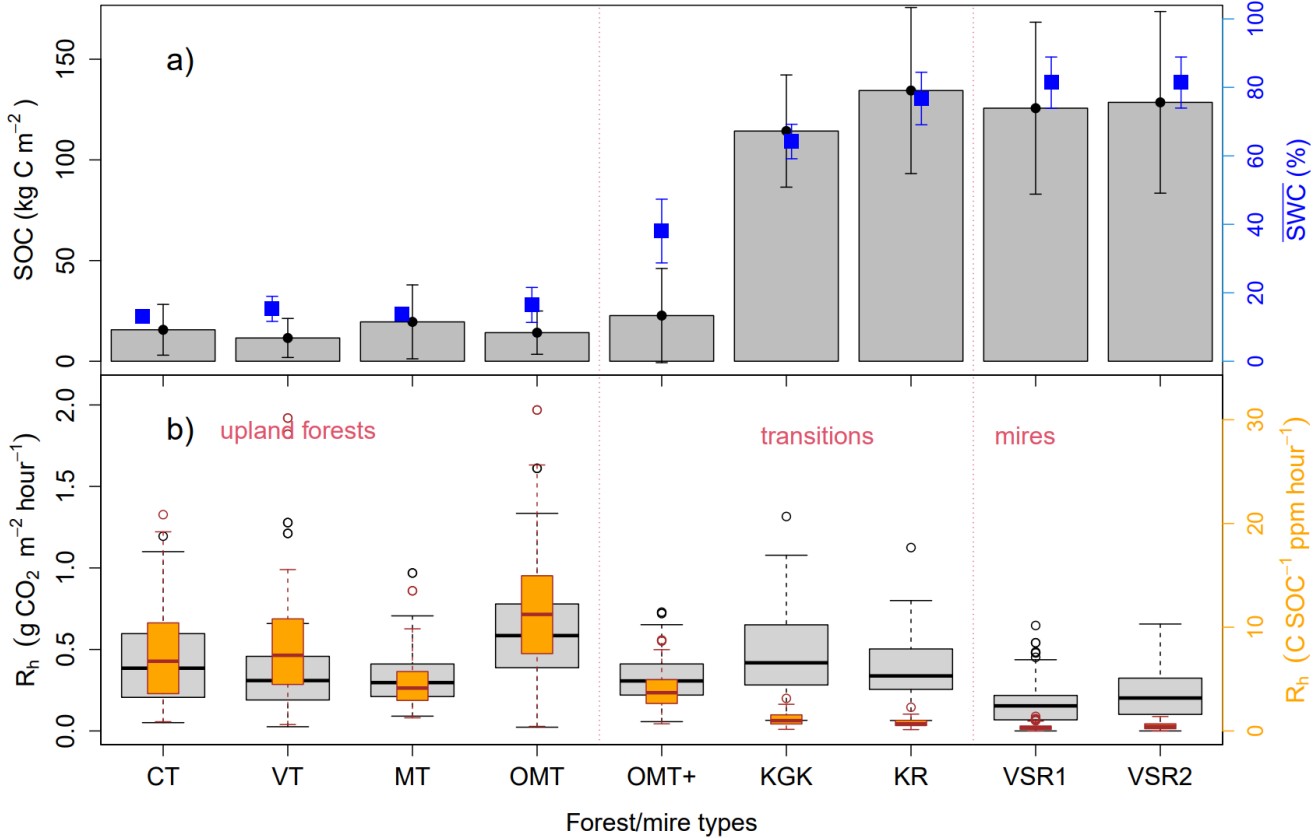

Figure 2. Forest/mire site type specific observations of soil organic carbon (SOC) stocks (kg C m$^{-2}$, summed up to 1 m) and the mean the volumetric soil water content (SWC) at 10 cm depth (%) (with error bars showing one standard deviation) (a) in comparison to their distributions of heterotrophic soil $CO_2$ emissions/respiration measurements (Rh, g$CO_2$ m$^{-2}$ h$^{-1}$) and R$_h$ expressed as the emitted C fraction per site specific SOC stock (C SOC$^{-1}$ ppm h$^{-1}$) (b). The CT, VT, MT, and OMT types represent upland forests, OMT+, KgK, and KR forest-mire transitions, and VSR1 and VSR2 mires. The boxplot horizontal lines show 25$^{th}$ and 75$^{th}$ interval with median in between, and 5$^{th}$ and 95$^{th}$ confidence interval (whiskers).




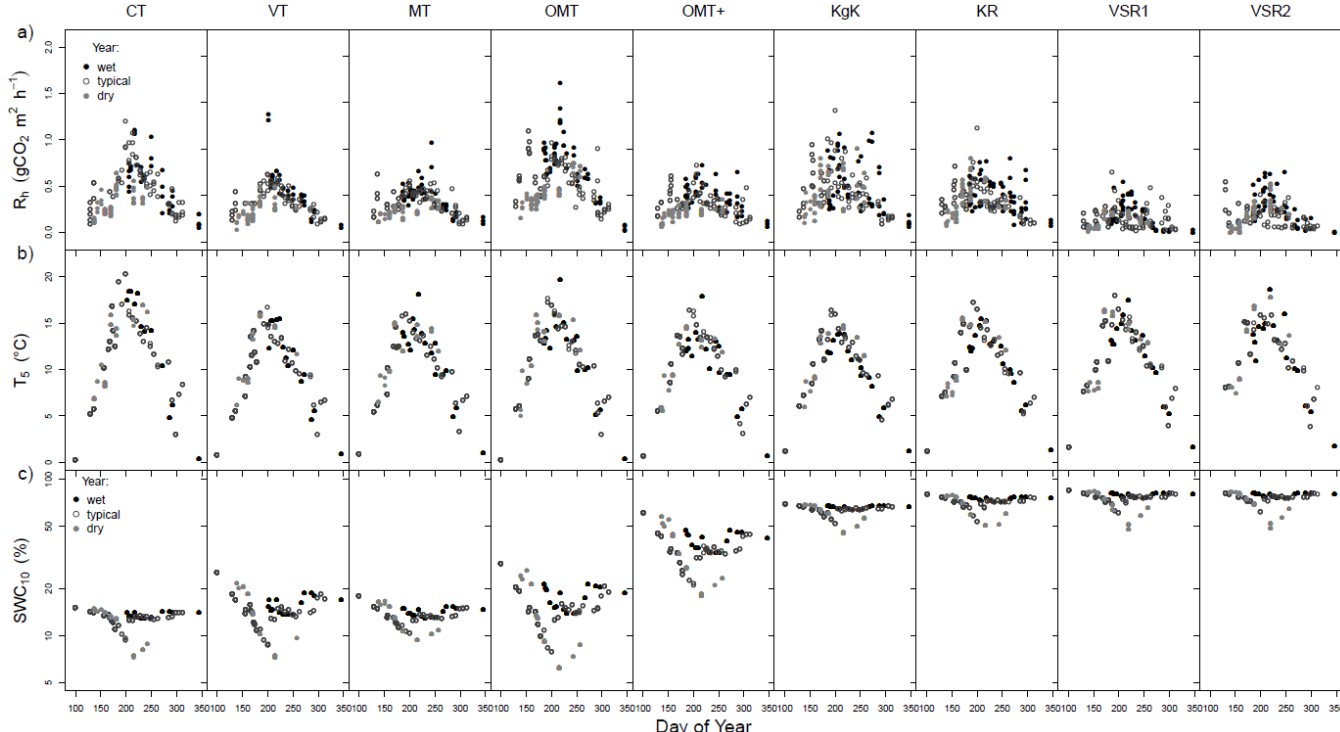

Figure 3. The three years' time series (2004 - wet, 2005 - typical, and 2006 - dry) of instantaneous measurements of a) soil
heterotrophic respiration ($R_h$, $gCO_2$ $m^{-2}$ $h^{-1}$, positive sign), b) soil temperature at 5 cm depth (°C), and c) soil moisture at 10 cm
depth (%)) of 9 forest/mire sites (4 upland forests (CT,MT, and OMT), 3 forest-mire transitions (OMT+, KgK, and KR) and 2
mires (VSR1 and VSR2). The sites are arranged from left to right according to their position on the slope (see Fig. 1).





Figure 4. The optimized environmental modifier $\xi_D$ of default decomposition rates (Eq. (4)) (coupled with Yasso07 model) drawn with mean posterior values of parameters (Table 2) for separate responses to (a) soil temperature at 5 cm, $\xi_D = f(T_5)$ when $f(SWC_{10}) = 1$, (b) to soil water content at 10 cm, $\xi_D = f(SWC_{10})$ when $f(T_5) = 1$, and (c, d) combined $\xi_D = f(T_5, SWC_{10})$ from the posterior distributions based on SOC-CO2 data (p ($\theta$ | SOC-CO2)) or only SOC (p ($\theta$ | SOC)). In the panels of combined $\xi_D$ (c, d) white circles show pairs of corresponding monthly means of $T_5$ and $SWC_{10}$, and the black circles show the annual $T_5$ and $SWC_{10}$ means for 9 forest/mires site types. The levels in colour gradients approximate the levels of the contour lines.



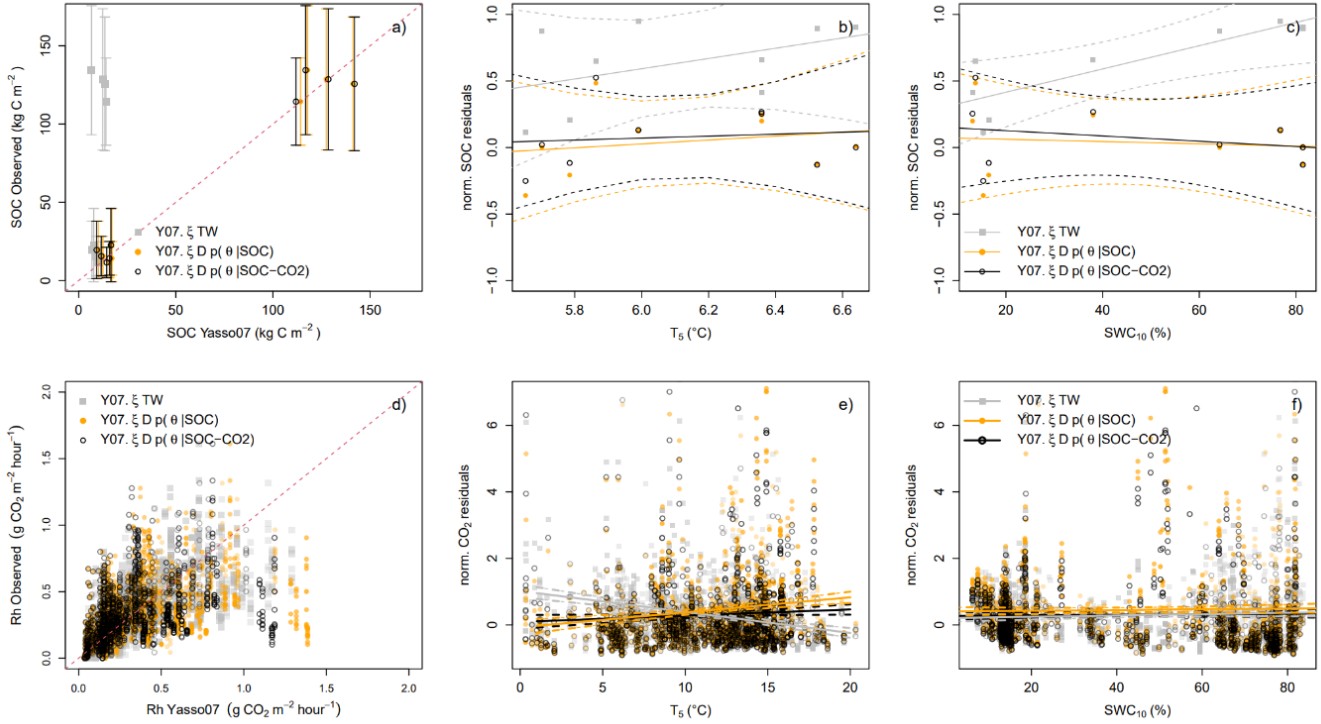

Figure 5. Scatterplots between observed SOC (kg C m$^{-2}$) and Rh (g CO$_2$ m$^{-2}$ hour$^{-1}$) from the forest-mire ecotone against modelled values with the two versions of Yasso07 model (i) Y07.ξ$_{TW}$ – Yasso07 coupled with the default environmental modifier (ξ$_T$, Eq. (3)) based on air T and precipitation with global parameter set (Tuomi et al., 2011) (applied for CT…KR mineral and organo-mineral soil forest sites) and with the reduction of decomposition rates by 65% for wetlands (Goll et al., 2015, Kleinen et al., 2021) (applied for VSR1, VSR2 mires sites), and (ii) Y07.ξ$_D$ - Yasso07 coupled with environmental modifier (ξ$_D$, Eq.3) based on soil T at 5 cm (T$_5$, °) and SWC at 10 cm depth (SWC$_{10}$, %) (Davidson et al., 2012) using posterior parameters of ξ$_D$ based on SOC (p (θ | SOC)) or based on SOC-CO2 data (p (θ | SOC-CO2)) (a, d) compared with 1:1 line (dashed red line). The model residuals normalized with the observations (norm. SOC and CO$_2$ residuals = residuals/observations) are plotted against the T$_5$ and SWC$_{10}$ with the trendlines of the linear fits and with their confident intervals (dashed lines) (b, c, e, and f).





Table 1. The parameters of the forest/mire site specific nonlinear regression models of soil heterotrophic respiration (Eq. (1))
with their root mean square error (RMSE) and degree of freedom (DF).

| No. | Forest/mire type | Rhref | $Q_{10}$ | SWCopt (%) | RMSE | DF |
|---|---|---|---|---|---|---|
| 1 | CT | 0.371 | 2.342 | 21.337 | 0.081 | 226 |
| 2 | VT | 0.290 | 2.407 | 18.252 | 0.066 | 244 |
| 3 | MT | 0.315 | 2.111 | 20.940 | 0.056 | 256 |
| 4 | OMT | 0.530 | 2.715 | 21.760 | 0.087 | 266 |
| 5 | OMT+ | 0.335 | 2.019 | 38.347 | 0.059 | 219 |
| 6 | KgK | 0.443 | 2.164 | 58.331 | 0.103 | 286 |
| 7 | KR | 0.375 | 1.772 | 63.486 | 0.068 | 303 |
| 8 | VSR1 | 0.157 | 1.621 | 67.154 | 0.042 | 300 |
| 9 | VSR2 | 0.225 | 1.929 | 66.264 | 0.065 | 245 |
| Ecotone | CT…VSR2 | 0.540 | 2.601 | 31.105 | 0.273 | 2369 |

Table 2. The posterior distribution of parameters of Yasso07 soil carbon model (parameters same as in Table S1) coupled with
environmental function $\xi_D$ (Eq. (4), parameters d, $Q_{10}$, $SWC_{opt}$) optimized with observations of SOC stocks (p ($\theta$ | SOC)) or
SOC stocks and $CO_2$ emissions (p ($\theta$ | SOC-CO2)) from forest/mire ecotone sites using Bayesian data assimilation (Hartig et
al., 2012). The PSRF stands for Gelman–Rubin potential scale reduction factor and MAP for a maximum a posteriori
probability.

| $\xi_D$ parameters | Posterior p ($\theta$ | SOC) | | | | | Posterior p ($\theta$ | SOC-$CO_2$) | | | | |
|---|---|---|---|---|---|---|---|---|---|---|
| | PSRF | MAP | 2.5% | 50% | 97.5% | PSRF | MAP | 2.5% | 50% | 97.5% |
| d | 1.028 | 0.9995 | 0.9990 | 0.9995 | 0.9999 | 1.048 | 0.9995 | 0.9992 | 0.9995 | 0.9995 |
| $Q_{10}$ | 1.014 | 4.338 | 1.262 | 4.047 | 4.964 | 1.019 | 3.425 | 2.513 | 3.56 | 4.4852 |
| $SWC_{opt}$ | 1.023 | 7.098 | 5.196 | 9.441 | 60.343 | 1.049 | 5.009 | 5.017 | 5.476 | 21.38 |




Table 3. The SOC and $CO_2$ performance statistics of Yasso07 (Y07) model versions against the measured data in boreal forest-mire ecotone where N is the number of observations, MAE is the mean absolute error, MBE is the mean bias error, RMSE is the root mean square error, $R^2_{adj}$ is the adjusted coefficient of determination, and AIC is the Akaike Information Criterion. The units of MAE, MBE, and RMSE are in kg C m$^{-2}$ and kg $CO_2$ m$^{-2}$ month$^{-1}$ for SOC and $CO_2$, respectively.

| Data | Yasso07 model | MBE | MAE | RMSE | $R^2_{adj}$ | AIC |
|---|---|---|---|---|---|---|
| SOC | Y07.$\xi_{TW}$ | -54.97 | 54.97 | 76.67 | 0.05 | 87.06 |
| (N = 9) | Y07.$\xi_{D,p\ (\theta\,\mid\,SOC)}$ | 0.20 | 6.92 | 9.06 | 0.97 | 67.84 |
| | Y07.$\xi_{D,p\ (\theta\,\mid\,SOC\text{-}CO2)}$ | -11.74 | 11.99 | 16.90 | 0.95 | 73.45 |
| $CO_2$ | Y07.$\xi_{TW}$ | 0.01 | 0.15 | 0.12 | 0.39 | -4137.83 |
| (N= 2644) | Y07.$\xi_{D,p\ (\theta\,\mid\,SOC)}$ | 0.05 | 0.21 | 0.16 | 0.15 | -3095.09 |
| | Y07.$\xi_{D,p\ (\theta\,\mid\,SOC\text{-}CO2)}$ | 0.00 | 0.19 | 0.12 | 0.16 | -3427.99 |