# Peer review of "Modeling boreal forest's mineral soil and peat C stock dynamics with Yasso07 model coupled with updated moisture modifier"

_EGUsphere, 2023_

## Referee Comment (RC1)

**Overall**

The study is relatively well described and fairly transparent. It deals with a very timely and important aspect, and I welcome the efforts of the authors to present a relatively simple soil moisture modifier for decomposition as estimated by the Yasso07 model. Thus, a relatively simple "solution" to a very complicated challenge.

In my reading of the paper I have not been able to see that you consider that the sites (to the best of my observations) have different soil water retention characteristics. And my most important comment is that I ask the authors to i) if they have done so then to describe this in much higher detail or ii) if they do not consider soil water retention characteristics – then I ask that it is considered specifically.

Further, I lack a number of details in the generation of the model input data and the field methodology which I ask is worked through more thoroughly than in the present version of the paper.

I also ask that the authors discuss more thoroughly several of the assumptions made and uncertainties and what effects they may have on the results.

(1)

The parameters they find for this modifier indicate that the optimal volumetric soil water content (SWC) at 10 cm depth is below 10%. For their sites (9 sites representing a gradient in long term moisture i.e. upland forest – transitional – mire) such conditions are only found, during the study, at the upland forest sites. I do not find in their descriptions that they consider the site specific soil water retention characteristics.

As I understand the consequences of the resulting modifier, a prediction of Rh in a mire that is drained may never encounter "optimal SWC conditions" as it may be physically impossible/unlikely for peat to reach such low volumetric soil contents (SWC of 10% may be dryer than wilting point in such soils, see fx. below). It seems counter intuitive that drained peat soils – during a phase of drainage and drying - should not pass through a stage of quite optimal conditions for decomposition. Therefore, I ask that the authors include more detailed considerations of how the differences in water retention characteristics (among site types/soil types) have been included in their considerations.

Perhaps including the findings and considerations found in Ghezzehei et al. 2019 are useful: "On the role of soil water retention characteristic on aerobic microbial respiration" Teamrat A. Ghezzehei, Benjamin Sulman, Chelsea L. Arnold, Nathaniel A. Bogie, and Asmeret Asefaw Berhe. https://bg.copernicus.org/articles/16/1187/2019/: "*Unless empirical moisture sensitivity curves are calibrated individually for each soil, ignoring the independent contributions of water potential and water content on microbial activity is tantamount to discounting the role of soil texture and structure on soil moisture sensitivity curves. This drawback is especially critical in land-surface models that might be applied across many different soil types.*"

Example (figure) of different soil water retention curves in different soils:

[Figure]

Figure 7. Soil water characteristic curves for four different soils. These curves show the relationship between the water content and the soil water potential.

Figure by: Soil and Soil Water Relationships: Zachary M. Easton, Assistant Professor and Extension Specialist, Biological Systems Engineering, Virginia Tech Emily Bock, Graduate Research Assistant, Biological Systems Engineering, Virginia Tech. Publication BSE-194P.

(2)

Yasso07 works on the basis of defining the chemical quality of litter assuming that this – with litter size, climate – controls the rate of decomposition. Across the sites in this study there will be large differences in the vegetation producing the litter entering the soil (and the model). In the paper I miss the values used for each litter type and plant species. This, I believe, should be available in the supplement. Fx Lang et al. 2009 (Journal of Ecology 2009, 97, 886–900) shows a factor 10 difference in the magnitude of the two-year litter mass loss in different species of sphagnum mosses. My expectation is that how the litter chemical characteristics across very different ecosystems (and plant species) are chosen, will influence the modifier found through the Bayesian method. I expect that AWEN (litter chemical characteristic) for tree litter to a large extent has been measured with some certainty, but I am less assured of the AWEN for ground vegetation species including sphagnum mosses. Given the potential wide use and thus high impact of such studies done here when it comes to global GHG and C modelling and in national C reporting and accounting to the UNFCCC , I recommend that the supplement is used to include as much information on method details as possible, also the AWEN used.

(3)

Root litter after trenching (flux chamber collars) enters the soil (and the model) in higher amounts in upland and transitional sites than in mire sites. I would like to ask the authors to describe to which degree this influences the outcome of the fitted parameters of the modifier.

(4)

The modifier is parameterized for two cases i) only using SOC stock data and ii) using Rh and SOC data. In the latter case a weighing is applied between the two data types. I would like to ask the authors why a modifier was not fitted using only Rh data and to what extent the weighing in ii) influenced the parameterization of the modifier.

(5)

I find that parts of the methods (field methods and upscaling of climatic variables fx) as well as the discussion needs to be worked through. Notes below indicate where I seem to lack information or find vague descriptions/sentences.

**Specific comments – line by line**

The graphical abstract. Here you show the modifier indicating it depends on relative soil moisture. These are terms not mentioned in the rest of the paper and not in equation 4 which describes the modifier and line 140 where you define SWC10. Please explain the relationship between SWC10 and "relative soil moisture". Also the term "Moisture index" is not used elsewhere in the paper in direct relationship with the modifier or with SWC10.

..during THE years..

   how many measurements for each site/plot? were the depth of instrument measured from the top of the forest floor or the top of the mineral soil? i.e. were instruments consistently placed in the forest floor (humus), in mineral soil or a mixture depending on the depth of the forest floor?

   you write SWC measured at depth of 10 cm, in line 144 you write that SWC of top 10 cm were….did you measure AT depth 10 cm or TO depth 10 cm? (in the latter case assuming from a defined level fx. 0-10 cm from the top of the forest floor??). in line 140 you define your variable as SWC10 however in line 144 you don't use the same term rather "SWC of top 10 cm" . do you mean the same thing or are these different variables? From where are the 10 cm depth measured? (top of forest floor? Top of mineral soil?). Do all SWC10 measurements represent similar positions at depth relative to forest floor (LFH horizon) and mineral soil horizon?

   "SWC of top 10 cm were in the same order of magnitude between the forest/mire site types". I assume you here mean that the lower row of figures in Figure 3 show that SWC10 for the 9 sites vary between a winter SWC10(vol%) of ca. 15% (driest upland sites) to ca. 60-70% (mire types). I don't understand why you describe this as a case where sites show SWC10 of "same order of magnitude"…please develop your argument for why you believe sites have similar SWC10?

gap-filling regressions did not include precipitation events at your site..why not?, not available?

so, you are calculating (for gap closing) monthly SWC10 and T5 from relationship with met data 6 km away. what do the regressions look like and how closely do data from the met site and the study site correlate for SWC10 and T5?

is this done with consideration of forest canopy conditions /seasons i.e. potential higher light (and rain water) interception at some time points than in others? does the met station data represent conditions of standard PET i.e. well watered grass? and does the forest gradually get dryer during the growing season i.e. this could influence the prediction of your monthly upscaled study-site SWC and T. please explain the gap closing in more detail incl. to what extent seasonal aspects influenced correlations between site data and met station data.

so 3 chamber measurement positions on each site?

remove "s" in "collars"

replace "clipped" by "removed" …

"half an hour"

along the…perimeter?

THE humuslayer

Finer et al. finds that in boreal forest fine roots below ca. 30 cm make up on average 20-30 %. Please discuss the potential effect on your results.

ground water level

168-169    please check (and include) the units.

please present/explain the spline function in a bit more detail.

breast height monthly values of T and SWC10?

where do precip and temp data come from?, the met station?

monthly data for T and SWC10? (please use SWC10 and T5 consistently instead of soil moisture if this is what you mean)

wording..the H pools does change, only very very slowly..or?

default meaning Tuomi et al. 2011?

check wording, something is missing.

i interpret your method to SWCopt = the SWC vol% at 10 cm at which Rh is optimal. thus - not a relative SWC normalized by fx field capacity or something like that. is that correct? this is because you later indicate that the optimal SWC is a SWC of 5-10%

why were not measured Rh scaled to monthly values instead? Please describe how.

please include a description/table or the like on how you did the monthly distribution of litter.

what is the magnitude and duration of the $CO_2$ emissions by the cut roots relative to the bulk soil, FF and other OM not affected by collar installation? Can you provide some sort of estimate for this?

just to be sure I understand: Evaluation against the data used for parametrization? Or was there data "left out" in order to do some leave-out validation/evaluation?

20 kgC/m2 seems rather high for Finland (Rantakari et al. 2012).

please give rationale for the chosen indicator please chose a wording or acronym throughout the paper to indicate if you are talking about the measured (instantaneous) values or the upscaled monthly values of soil water content (and temperature).

I am unsure in which context you mean the "optimum" here please confirm if this is SWC in volumetric % or if these should be understood as some kind of normalized values of SWC?

use among, not between something is wrong with the wording.

…spatially prevailing..i do not understand. Please revise wording.

..wider and higher increase…revise wording.

..THE two Yasso…did not INDICATE any bias relative to…

..THE original yasso…

complicated sentence. Revise. Do you mean "however" instead of "although"...?

369-372      language. Please revise.

language. move "empirical" to before "soil".

I assume that you mean that the general method of using Bayesian MCMC has proven useful in other studies with other data?, please confirm and adjust wording accordingly.

response curve....specify the response of what to what have a look at results in https://bg.copernicus.org/articles/16/1187/2019/

here you list different SWCoptima for soil of different soil properties. But your result in the modifier indicates a SWCoptima of <10vol. % for all of your 9 sites?, please confirm that I understand your conclusion correct. But at the same time you recognize that optima should logically differ according to soil properties. I am not convinced by your documentation that your 9 sites have similar soil properties.

language. Functions or equations do not impact decomposition. Please revise so sentence reflect what you mean.

do you mean here that table 1 shows that the Rh-ref was highest in upland forest?...

high legacy field soil moisture. What do you mean by this?

if results in Das et al. is relevant for your results then please give a description of what they find and how it adds to the understanding of your results.

what is meant by "field moisture"?, please use your own defined variable names consistently if that is what you mean.

would be nice to see the site specific effect of the trenching i.e. where a large amount of fresh litter is added to the plot and unevenly distributed among site types. Please indicate this in the appropriate figures fx.

how would your model with new modifier predict a case of relatively gradual drainage of mires where the surface SWC changes from say 80% to 60% over a few years/months.? would Rh be reduced to 20% of its potential (figure 4b) at SWC of 60 vol%? Even if - depending on the site specific soil water retention characteristics - the drained site would now have a moisture regime most often mid between field capacity and wilting point.

for SOC i would agree. for CO2 i see indication that model is overestimating Rh. 5d: low observed Rh in many cases modeled as high Rh values in Yasso.

yes, i think so too. and you do a weighing of your data when used in the fitting of the Bayesian routine, right? what is the sensitivity of results if this weighing was to be changed between the two data types?

redundant language: increase is very common from lower to higher values...

something is missing, language.

yes, you measure Rh from soil deeper than 10 cm but do not account in your SWC measurements for the moisture in the deeper soil which most likely will depend on both site/soil type and season (weather, evaporative demand, root distribution etc). Please quantify/discuss the level of uncertainty this will cause in your input to yasso and the modifier.

do you mean microbial growth respiration? Or: microbial growth or microbial respiration?

add a "availability" by the end of the sentence.

---

## Author Response (AR1)

Egusphere-2023-1523

**Reply to reviewers on "Modeling boreal forest's mineral soil and peat C dynamics with Yasso07 model coupled with Ricker moisture modifier"**

**Tupek et al.** boris.tupek@luke.fi

We thank both reviewers for thoughtful and insightful evaluation of our study, and for constructive comments which helped to improve the paper! Our replies are highlighted in yellow, or green when referring to the implementation of the comments in the revised paper.

**General replies and major improvements in the revised paper include:**

1) in reply to the comments on <10% SWC optimum for decomposition

To address probably the main reason for the SWCopt <10% we re-designed the functional form of dependency of decomposition to soil water content (SWC) to better account for the reduction of respiration towards zero SWC by using a modified Ricker function (instead of previously used Gaussian which was biased for dry soils and thus also forced SWC optimum to lower values).

The hump-shaped Ricker function has specific ascending/descending slopes (Bolker, 2008). We adjusted it for application as a moisture modifier by scaling it to 1 due to its combination with the functional dependency for temperature in the environmental modifier of the soil C models (explained in detail in revised methods). The newly formulated moisture function could be controlled and calibrated just by one parameter thus making it more theoretically sound, robust, and applicable for the soil C models.

The Ricker function improved the representation of decomposition for drier soils and the representation of optimal SWC for decomposition. The SWC optimum was derived from the fitted ascending slope parameter (equation in the methods). The SWC optimum values found with the Ricker function were between 14 and 27 % (depending on the data used in MCMC) which matched well with the observed SWC conditions of well drained mineral soils in boreal forest (Figure 2 in the preprint).

2) in reply to comments on weighting $CO_2$ error in MCMC calibration

We deleted the weighting of $CO_2$ errors in the likelihood. With the Ricker moisture function, it was also unnecessary to use least-square regression (NLS) for informing the priors for MCMC. Thus, we also deleted NLS part from the paper.

3) As requested, we included MCMC with $CO_2$ for comparison to SOC and SOCCO2

4) As requested, we validated the estimated parameters by separating data for fitting the models and testing with 9-fold cross validation technique.

**Detailed replies to specific comments and their improvements:**
**Referee#1**

**Overall**

The study is relatively well described and fairly transparent. It deals with a very timely and

important aspect, and I welcome the efforts of the authors to present a relatively simple soil moisture modifier for decomposition as estimated by the Yasso07 model. Thus, a relatively simple "solution" to a very complicated challenge.

Thank you!

In my reading of the paper I have not been able to see that you consider that the sites (to the best of my observations) have different soil water retention characteristics. And my most important comment is that I ask the authors to i) if they have done so then to describe this in much higher detail or ii) if they do not consider soil water retention characteristics – then I ask that it is considered specifically.

In the revised version of the manuscript, we clarified our emphasis on the topsoil humus layer to enable the applicability of the moisture modifier across different soil types.

It was clear that moisture of the deeper soil horizons cannot be used for fitting a common function (for the same concern as yours that the mineral soils water retention varies, and these are very different from the peat). Developing one functional form for net soil $CO_2$ emission and SOC stocks among the forest mire types could be done only for topsoil humus layer because of similar properties across the soil types e.g., such as porosity, bulk density, soil water retention. This was briefly mentioned in the preprint (lines 140-145) and agrees with Launiainen et al. (2022) who studied water retention for topsoil humus layer in boreal forest.

Launiainen, S., Kieloaho, A.-J., Lindroos, A.-J., Salmivaara, A., Ilvesniemi, H., Heiskanen, J., 2022. Water Retention Characteristics of Mineral Forest Soils in Finland: Impacts for Modeling Soil Moisture. Forests 13, 1797. https://doi.org/10.3390/f13111797

Further, I lack a number of details in the generation of the model input data and the field methodology which I ask is worked through more thoroughly than in the present version of the paper.

I also ask that the authors discuss more thoroughly several of the assumptions made and uncertainties and what effects they may have on the results.

(1)

The parameters they find for this modifier indicate that the optimal volumetric soil water content (SWC) at 10 cm depth is below 10%. For their sites (9 sites representing a gradient in long term moisture i.e. upland forest – transitional – mire) such conditions are only found, during the study, at the upland forest sites. I do not find in their descriptions that they consider the site specific soil water retention characteristics.

As already mentioned in the discussion of the preprint (lines 463-474) the SWC10 optimum < 10% was partly 1) an artefact of the Gaussian function which does not allow the ascending and descending slopes to vary and 2) relatively few data for extremely low SWC. This was improved in the revised analysis by using modified Ricker function (replacing Gaussian).

The exact reason for SW optimum of decomposition found in well-drained soils could be evaluated in more detail in future studies e.g., by comparing performance of deterministic and mechanistic functions. However, our results already contribute to the advancement of soil C modeling and provide insights into reducing $CO_2$ emissions from managed organic soils through adjusting topsoil SWC via water level management.

As I understand the consequences of the resulting modifier, a prediction of Rh in a mire that is drained may never encounter "optimal SWC conditions" as it may be physically impossible/unlikely for peat to reach such low volumetric soil contents (SWC of 10% may be dryer than wilting point in such soils, see fx. below).

It seems counter intuitive that drained peat soils – during a phase of drainage and drying - should not pass through a stage of quite optimal conditions for decomposition. Therefore, I ask that the authors include more detailed considerations of how the differences in water retention characteristics (among site types/soil types) have been included in their considerations.

In revised paper with Ricker function the boreal forest mire ecotone SWC optimum has been found at around 15-23% of **SWC10 (SWC in topsoil and NOT in the deeper peat)**. Around 20% SWC for topsoil humus is common moisture condition with wilting point for humus at 11 % (Launiainen et al. 2022). So, in comparison to Figure 7 (that was given as an example) topsoil humus and typical peat have slightly different water retention curves. Fitting the SWC function based on the topsoil humus, as a proxy correlating the environmental conditions in the landscape to soil C stock and heterotrophic $CO_2$ emissions, was one of the best solutions here. The use of topsoil humus SWC does not necessarily require including properties of deeper soils (e.g., use of their water retention curves).

Regarding decomposition of drained peat soils, the function informs that until the top layer has reached 20 % SWC10 the mean decomposition rate of the peat down to 1 m is indeed not at the fastest rate. Whether the deterministic modifier rate was estimated correctly or not also for the drained peatlands should be tested in follow up studies, as our data did not include drained peatlands. The Ricker functional dependency has performed better for the drier region but the performance in soils with high water status still could be improved. This could be deduced from better statistical performance of $CO_2$ only fit with $CO_2$ data (compared to SOC or $SOCCO_2$ fit) which produced larger tail of the Ricker function. Although, the $CO_2$ only fit also underestimated SOC stocks of forested peatlands. So, the scale sensitivity between the SOC vs $CO_2$ based functions could be also evaluated.

[Figure]

Figure 7. Soil water characteristic curves for four different soils. These curves show the relationship between the water content and the soil water potential.

Example (figure) of different soil water retention curves in different soils:

Figure by: Soil and Soil Water Relationships: Zachary M. Easton, Assistant Professor and Extension Specialist, Biological Systems Engineering, Virginia Tech Emily Bock, Graduate Research Assistant, Biological Systems Engineering, Virginia Tech. Publication BSE-194P.

Perhaps including the findings and considerations found in Ghezzehei et al. 2019 are useful: "On the role of soil water retention characteristic on aerobic microbial respiration" Teamrat A. Ghezzehei, Benjamin Sulman, Chelsea L. Arnold, Nathaniel A. Bogie, and Asmeret Asefaw Berhe. https://bg.copernicus.org/articles/16/1187/2019/: "*Unless empirical moisture sensitivity curves are calibrated individually for each soil, ignoring the independent contributions of water potential and water content on microbial activity is tantamount to discounting the role of soil texture and structure on soil moisture sensitivity curves. This drawback is especially critical in land-surface models that might be applied across many different soil types.*"

We added following text into discussion:

Ghezzehei et al. (2019) suggested that empirical moisture sensitivity curves should be calibrated individually for each soil. However, our study shows that the common modifier function, based on the SWC of the topsoil humus layer only which has comparable properties across the soil types, could provide insights into a more generalizable moisture sensitivity function.

The mechanistic diffusion-based moisture functions e.g., by Ghezzehei et al. (2019) could be in follow up studies compared against deterministic moisture functions (e.g., as in Davidson et al. 2012) to evaluate their applicability and interpretation.

(2)

Yasso07 works on the basis of defining the chemical quality of litter assuming that this – with litter size, climate – controls the rate of decomposition. Across the sites in this study there will be large differences in the vegetation producing the litter entering the soil (and the model). In the paper I miss the values used for each litter type and plant species. This, I believe, should be available in the supplement. Fx Lang et al. 2009 (Journal of Ecology 2009, 97, 886–900) shows a factor 10 difference in the magnitude of the two-year litter mass loss in different species of sphagnum mosses. My expectation is that how the litter chemical characteristics across very different ecosystems (and plant species) are chosen, will influence the modifier found through the Bayesian method. I expect that AWEN (litter chemical characteristic) for tree litter to a large extent has been measured with some certainty, but I am less assured of the AWEN for ground vegetation species including sphagnum mosses. Given the potential wide use and thus high impact of such studies done here when it comes to global GHG and C modelling and in national C reporting and accounting to the UNFCCC , I recommend that the supplement is used to include as much information on method details as possible, also the AWEN used.

We added the corresponding table into the supplement. This includes the AWEN values for different species and their components, used for litter input modelling into the text of the paper.

(3)

Root litter after trenching (flux chamber collars) enters the soil (and the model) in higher

amounts in upland and transitional sites than in mire sites. I would like to ask the authors to describe to which degree this influences the outcome of the fitted parameters of the modifier.

Yes, the root litter after the trenching initially enhances soil $CO_2$ emission and stabilizes within few weeks or months depending on the quality and quantity of the decomposing roots. This was originally accounted for in the modeling of the litter input with the peak after trenching (see Figure S3) which subsequently produced increased $CO_2$ emissions from the soil C model.

This was included into discussion of the revised paper as requested.

(4)

The modifier is parameterized for two cases i) only using SOC stock data and ii) using Rh and SOC data. In the latter case a weighing is applied between the two data types. I would like to ask the authors why a modifier was not fitted using only Rh data and to what extent the weighing in ii) influenced the parameterization of the modifier.

In the revised version of the analysis: 1) we do not use weighting of the error terms anymore as the error term in the likelihood removes the differences among the datasets. Using Ricker functional form for moisture dependency improved estimates of $CO_2$ emissions in drier soils and the models were converging quickly regardless the data source; 2) we also included MCMC parametrization based only on soil $CO_2$ data for comparison with SOC and SOC-$CO_2$ approaches.

(5)

I find that parts of the methods (field methods and upscaling of climatic variables fx) as well as the discussion needs to be worked through. Notes below indicate where I seem to lack information or find vague descriptions/sentences.

**Specific comments – line by line**

50      The graphical abstract. Here you show the modifier indicating it depends on relative soil moisture. These are terms not mentioned in the rest of the paper and not in equation 4 which describes the modifier and line 140 where you define SWC10. Please explain the relationship between SWC10 and "relative soil moisture". Also the term "Moisture index" is not used elsewhere in the paper in direct relationship with the modifier or with SWC10.

We clarified in graphical abstract legend: f(SWC/porosity).

By relative water content we meant = SWC/porosity which was used comparison with the functions used in different soil C models. Moisture index in the figure means SWC/porosity in range from 0 to 1 .

129     ..during THE years.. implemented

140            how many measurements for each site/plot? were the depth of instrument measured from the top of the forest floor or the top of the mineral soil? i.e. were

instruments consistently placed in the forest floor (humus), in mineral soil or a mixture depending on the depth of the forest floor?

this was explained in detail in given references to Tupek et al. 2008, 2015

141         you write SWC measured at depth of 10 cm, in line 144 you write that SWC of top 10 cm were….did you measure AT depth 10 cm or TO depth 10 cm? (in the latter case assuming from a defined level fx. 0-10 cm from the top of the forest floor??). in line 140 you define your variable as SWC10 however in line 144 you don't use the same term rather "SWC of top 10 cm" . do you mean the same thing or are these different variables? From where are the 10 cm depth measured? (top of forest floor? Top of mineral soil?). Do all SWC10 measurements represent similar positions at depth relative to forest floor (LFH horizon) and mineral soil horizon?

We clarified that by SWC10 we mean SWC of top soil moss+litter+humus layer. The exact soil surface is difficult to define in the field though the measurements of the moisture sensors represented moisture conditions of the decomposed part of litter in the humus horizon.

144         "SWC of top 10 cm were in the same order of magnitude between the forest/mire site types". I assume you here mean that the lower row of figures in Figure 3 show that SWC10 for the 9 sites vary between a winter SWC10(vol%) of ca. 15% (driest upland sites) to ca. 60-70% (mire types). I don't understand why you describe this as a case where sites show SWC10 of "same order of magnitude"…please develop your argument for why you believe sites have similar SWC10?

We clarified that:

The SWC10 values among the forest and mire site types ranged between 0 and 1 (or 0 and 100 %) (Figure 3), whereas in comparison to water level depth the values range from 8 cm to 881 cm (Tupek et al. 2008, Table 1).

146         gap-filling regressions did not include precipitation events at your site..why not?, not available?

This part was revised in the paper.

For missing field campaigns during months with the snow cover (Nov 2016, Feb – Apr 2005, Dec 2005 – Apr 2006) we interpolated the measured monthly mean T5 and SWC10 time series with a spline function.

148         so, you are calculating (for gap closing) monthly SWC10 and T5 from relationship with met data 6 km away. what do the regressions look like and how closely do data from the met site and the study site correlate for SWC10 and T5?

is this done with consideration of forest canopy conditions /seasons i.e. potential higher light (and rain water) interception at some time points than in others? does the met station data represent conditions of standard PET i.e. well watered grass? and does the forest gradually get dryer during the growing season i.e. this could influence the prediction of your monthly upscaled study-site SWC and T. please explain the gap closing in more detail incl. to what extent seasonal aspects influenced correlations between site data and met station data.

We clarified that monthly T5 and SWC10 and gap filling was needed only for winter months with the snow cover when the variation of soil climate data is minimal, and thus in revised paper we predicted them from the time series of the measurements by interpolation.

These **interpolated T5 and SWC10 values** were needed only for running Yasso07 in time series but **had realistically zero impact on the calibration** of the environmental functions, as these $CO_2$ model outputs were missing measured counterparts and thus not entering the MCMC data assimilation.

Linear regressions used for upscaling of hourly values to monthly level were based on the match between hourly measured T and SWC values on 9 stations of our forest – mire transect on the continuously hourly values measured at SII station. The $R^2$ coefficients were above 0.9.

| | |
|---|---|
| 152 | so 3 chamber measurement positions on each site? **Text** was **revised accordingly** (TRA) |
| 155 | remove "s" in "collars" TRA |
| 156 | replace "clipped" by "removed" … TRA |
| 156 | "half an hour" TRA |
| 157 | along the…perimeter? TRA |
| 159 | THE humuslayer TRA |
| 159 | Finer et al. finds that in boreal forest fine roots below ca. 30 cm make up on average 20-30 %. Please discuss the potential effect on your results. added into discussion |
| 161 | ground water level TRA |
| 168-169 | please check (and include) the units. TRA |
| 169 | please present/explain the spline function in a bit more detail. It is presented in detail in Fig. S1 |
| 176 | breast height TRA |
| 188 | monthly values of T and SWC10? TRA instantenous |
| 203 | where do precip and temp data come from?, the met station? Reformulated: Temperature and precipitation data was from the nearest Finnish meteorological institute weather station located 3 km away from our study sites. |
| 204 | monthly data for T and SWC10? (please use SWC10 and T5 consistently instead of soil moisture if this is what you mean) TRA |
| 209 | wording..the H pools does change, only very very slowly..or? TRA |
| 211 | default meaning Tuomi et al. 2011? |

216          check wording, something is missing. TRA

228          i interpret your method to SWCopt = the SWC vol% at 10 cm at which Rh is optimal. thus - not a relative SWC normalized by fx field capacity or something like that. is that correct? this is because you later indicate that the optimal SWC is a SWC of 5-10% Yes, by SWCopt we mean the SWC10 value which is non limiting decomposition at which Rh is optimal. TRA

248   why were not measured Rh scaled to monthly values instead? Please describe how. Yes, in revised analysis the measured Rh were scaled to monthly means. TRA

258          please include a description/table or the like on how you did the monthly distribution of litter. Monthly litter distribution is shown in the supplement Figures S2 and S3.

262          what is the magnitude and duration of the CO2 emissions by the cut roots relative to the bulk soil, FF and other OM not affected by collar installation? Can you provide some sort of estimate for this? soil CO2 emission and stabilizes within few weeks or month (see reply to this on page 3 -4)

295          just to be sure I understand: Evaluation against the data used for parametrization? Or was there data "left out" in order to do some leave-out validation/evaluation?

As requested, we validated the estimated parameters by separating data for fitting the models and testing with 9-fold cross validation technique.

301          20 kgC/m2 seems rather high for Finland (Rantakari et al. 2012).

Yes, it is rather high but within a realistic range.

307          please give rationale for the chosen indicator This was complicated and deleted

319          please chose a wording or acronym throughout the paper to indicate if you are talking about the measured (instantaneous) values or the upscaled monthly values of soil water content (and temperature). TRA

320          I am unsure in which context you mean the "optimum" here TRA

323          please confirm if this is SWC in volumetric % or if these should be understood as some kind of normalized values of SWC? SWC is volumetric % TRA

324          use among, not between TRA

325          something is wrong with the wording. TRA

326          ...spatially prevailing..i do not understand. Please revise wording. TRA

345          ..wider and higher increase...revise wording. TRA

363          ..THE two Yasso...did not INDICATE any bias relative to... TRA

364        ..THE original yasso…TRA

367         complicated sentence. Revise. Do you mean "however" instead of "although"…? TRA

369-372         language. Please revise. TRA

375         language. move "empirical" to before "soil". TRA

377         I assume that you mean that the general method of using Bayesian MCMC has proven useful in other studies with other data?, please confirm and adjust wording accordingly. TRA

399         response curve….specify the response of what to what TRA

401         have a look at results in https://bg.copernicus.org/articles/16/1187/2019/ TRA

403         here you list different SWCoptima for soil of different soil properties. But your result in the modifier indicates a SWCoptima of <10vol. % for all of your 9 sites?, please confirm that I understand your conclusion correct. But at the same time you recognize that optima should logically differ according to soil properties. I am not convinced by your documentation that your 9 sites have similar soil properties.

The topsoil humus layer properties e.g., bulk density, porosity, SWC are comparable between the forest/mire types (Tupek et al. 2016, Launiainen et al. 2022).

The individual deeper soil horizons among the forest/mire types have different properties. However, our focus was on using common soil properties in the forest-mire ecotone. Individual models for the forest types should be subject of another paper. For clarity on the spatial upscaling at the forest-mire ecotone level, unnecessary NLS based forest type models were deleted.

407         language. Functions or equations do not impact decomposition. Please revise so sentence reflect what you mean. TRA

408         do you mean here that table 1 shows that the Rh-ref was highest in upland forest?... this was deleted TRA

417         high legacy field soil moisture. What do you mean by this? mean long term moisture TRA

418         if results in Das et al. is relevant for your results then please give a description of what they find and how it adds to the understanding of your results. TRA

419         what is meant by "field moisture"?, please use your own defined variable names consistently if that is what you mean. TRA

425         would be nice to see the site specific effect of the trenching i.e. where a large amount of fresh litter is added to the plot and unevenly distributed among site types. Please indicate this in the appropriate figures fx. Please see Figures S2 and S3, trenching affected only the fineroot and the coarse root litter. TRA

445         how would your model with new modifier predict a case of relatively gradual drainage of mires where the surface SWC changes from say 80% to 60% over a few years/months.? would Rh be reduced to 20% of its potential (figure 4b) at SWC of 60 vol%? Even if - depending on the site specific soil water retention characteristics - the drained site would now have a moisture regime most often mid between field capacity and wilting point.

Text in results was revised:

According to the fitted moisture modifier the **Rh rate in peatlands would be increased (not reduced) by 10% if the SWC10 changes from 80% to 60% (Figure 4b).** Increase in modifier value means increase in decomposition rate with max at 1, and vice versa. The function shows that if the drainage continues efficiently further (e.g., if drainage is combined with increased evapotranspiration) the maximum Rh rate of organic matter decomposition is reached when SWC10 would be around 20 %, otherwise it is limited by SWC. The exact value of Rh rate also depends on the temperature, as the final Rh rate is a combined T5 and SWC10 dependency (Figure 5). The Rh rates of this model prediction should be validated in follow up studies with data from drained forested peatlands (e.g., SOC change derived from peat subsidence rate, and soil CO2 emissions). This was also explained in reply to comment (1).

[Figure]

Figure 4. The optimized environmental modifier of default decomposition rates $\xi_{AR}$ (Eq. (3)) (coupled with Yasso07 model) drawn with mean posterior values of parameters and their confident intervals (dashed lines) (Table 1) for separate responses to (a) soil temperature at 5 cm, $\xi_{AR} = f(T_5)$ when $f(SWC_{10}) = 1$, (b) to soil water content at 10 cm, $\xi_{AR} = f(SWC_{10})$ when $f(T_5) = 1$. The functions were fitted based on $CO_2$, SOC, and combined $CO_2$ and SOC data.

[Figure]

Figure 5. The optimized environmental modifier of default decomposition rates $\xi_D$ (Eq. (3)) (coupled with Yasso07 model) drawn with mean posterior values of parameters (Table 1) for combined responses to soil temperature at 5 cm, $\xi_{AR} = f(T_5)$ and to soil water content at 10 cm, $\xi_D = f(SWC_{10})$ based on only SOC (a), SOC and $CO_2$ (b), or only $CO_2$ (c) data. In the panels of combined $\xi_{AR}$ white circles show pairs of corresponding monthly means of $T_5$ and $SWC_{10}$, and the black circles show the annual $T_5$ and $SWC_{10}$ means for 9 forest/mires site types.

446         for SOC i would agree. for CO2 i see indication that model is overestimating Rh. 5d: low observed Rh in many cases modeled as high Rh values in Yasso. TRA

449         yes, i think so too. and you do a weighing of your data when used in the fitting of the Bayesian routine, right? what is the sensitivity of results if this weighing was to be changed between the two data types? The weighting of data in the likelihood was removed.

457         redundant language: increase is very common from lower to higher values... TRA

458         something is missing, language. TRA

468         yes, you measure Rh from soil deeper than 10 cm but do not account in your SWC measurements for the moisture in the deeper soil which most likely will depend on both site/soil type and season (weather, evaporative demand, root distribution etc). Please quantify/discuss the level of uncertainty this will cause in your input to yasso and the modifier. The problem regarding CO2 emissions from dry mineral soils has been solved with the Ricker function. However, most CO2 emissions originate from the topsoil, and T5 and SWC10 in the topsoil dynamics are highly correlated with the deeper layers.

469         do you mean microbial growth respiration? Or: microbial growth or microbial respiration? microbial respiration TRA

477         add a "availability" by the end of the sentence. TRA

**Referee#2**

The manuscript presents an updated environmental modifier for the soil organic matter decay function within Yasso07 model. The authors use the field measurements of soil organic carbon (SOC) and soil heterotrophic respiration in an upland-peatland complex in southern Finland to calibrate the environmental modifier function, which simulates the effects of soil temperature and moisture on the organic matter decay. The study is timely and well-suited for publication in the GMD following authors' addressing the comments and suggestions outlined below.

Thank you! We have thoughtfully addressed all the outlined comments and suggestions in general reply to major comments (at the beginning of our reply), and as individual replies (below).

It was somewhat surprising to see the parameters in Tables 1 and 2 not aligning very well. Q10's estimated with the data assimilation using SOC and SOC+CO2 flux were much higher compared with those estimated using NLS approach using the heterotrophic CO2 flux alone and SWCopt were much lower.

There might be several reasons for the discrepancies between fitted parameters with NLS and MCMC approach. One of them is that in NLS parameters are centered around the starting values, unlike MCMC which is using maximum likelihood estimation informed by the prior distribution. Also, the sampling algorithms of parameters differ between NLS and MCMC. This could be to some extent fixed by using e.g., nls.multstart package. However, we decided to remove NLS analysis in the revised version as the message here should not be evaluating different fitting methods and NLS was not necessary anymore to inform about the parameters of the Ricker function for MCMC. We also think NLS model evaluation at the level of individual forest/mire types should be removed as the focus of the paper is mainly on the spatially larger forest-mire level data and the common function at the boreal forest level. We decided to move the evaluation of deterministic and mechanistic moisture dependencies with more emphasis on forest type soil differences to our follow up studies.

However, to address probably the main reason for the SWCopt <10% we re-designed the functional dependency of decomposition to soil water content (SWC) to better account for the reduction of respiration towards zero SWC by using a modified Ricker function (compared to Gaussian function used in preprint).

The Ricker function improved the representation of decomposition for drier soils and the representation of optimal SWC for decomposition. The SWC optimum was derived from the fitted ascending slope parameter and its values were between 14 and 27 % (depending on the data used for fitting; 14% for both SOC, SOCCO2 and 27% for CO2).

The MCMC fit with CO2 data produced larger SWCopt and larger tail in the Ricker function (compared to SOC or SOCCO2 fit). However, the CO2 only fit also underestimated SOC stocks of forested peatlands. Thus, the main reason for previously observed mismatch in SWCopt between NLS CO2 and MCMC SOC and SOCCO2 was the forcing data and the function used for calibration. When we used the same method (only MCMC) and the better function (Ricker with min in 0 for 0 SWC), we have still seen larger SWCopt for CO2 only fit; slightly better CO2 estimates but underestimated SOC stocks because the decomposition rates in highly water saturated conditions were not reduced enough. The Ricker functional dependency has performed well for the drier mineral soils and with SOC included in fitting also performed reasonably for peatlands. However, the follow up studies in soils with high water status will

need more data from (drained) forested peatlands and evaluation of various (deterministic and mechanistic) functional dependencies.

Authors attributed the difference to different basal respiration rates in Yasso07 and NLS, however I would argue that Q10 and SWCopt control the curvature the respiration's curve, and basal respiration rate, being a scaling parameter, should not affect the values of Q10 and SWC opt in such a profound way.

The basal respiration rates in Yasso07 and NLS do not differ just in the scale, but they are inherently different; one dynamically changing in time and the other constant. The basal respiration in Yasso07 is dynamically changing with the change in SOC stock and litter input, whereas in NLS the basal respiration is a constant parameter estimated just from the soil Rh. This could be clarified but we rather removed NLS from the paper to streamline the main message.

I think such a difference in these parameter values may be attributed to the weighing of the observations in the data assimilation approach: SOC stocks appear to have a much larger influence on the posterior parameters compared to the CO2 flux.

This is correct, weighting of observation towards SOC in the previous version of analysis could have such impact on estimated parameters.

In revised analysis we removed weighting from the likelihood.

The posterior values of parameters a and b from the equation 6 were not reported (I assume they were estimated, because there are prior values reported in the Table S1), so it was impossible for me to evaluate whether that may have been the case.

In revised analysis we report the posterior a and b error parameters.

I would suggest doing one more calibration experiment to explore whether the data weighing is an issue and calibrate the Yasso07 with CO2 observations alone. If the parameters are similar to the NLS, the weighing of the observations is likely the culprit. If it is, I would suggest weighing the observations by their individual errors: the larger is the error associated with the observation's mean value, the lesser weight it should be attributed within the calibration algorithm. This way the algorithm would not "hack" itself to produce the smallest error, but rather be forced to gain information from the more precise observations.

Yes, we included MCMC with CO2, and with the Ricker function and the likelihood without weighting it worked well.

For the clarity of the paper, it is indeed better to do one more MCMC with CO2 for comparison to MCMC SOC and SOCCO2 (than evaluate comparison between NLS CO2 and MCMC SOC and SOCCO2).

It was not clear from the methods whether the data used for model validation were the same as the data used for model calibration. If the data were the same, I would suggest re-doing the calibration with less observations and reserving a portion of the observations for model validation. If the two observation sets are already separated, please include this information in the methods section.

In revised analysis we validated the estimated parameters by separating data for fitting the

==models and testing with 9-fold cross validation technique.==

The units of respiration are in g CO2, and within the model the units associated with C transfers are in g C, were the units of respiration converted to gC before calibration?

==Yes, this was double checked. The correct units are used.==

Below is the list of the minor comments and suggestions:

L23-24: " "…calibrated against SOC and CO2 data using Bayesian MCMC approach showed ….." **Text** was **revised accordingly** (TRA)

L70: "…in underestimation…" TRA

L55: what metal are the collars made of? Can it affect respiration rate? ==it was stainless steel collar, and we think it had a minor effect on soil respiration==

L66-67: please include the depth increments ==These can be seen in Fig. S1.== TRA

L176: "Breast height…" TRA

L214: if inputs and pools are functions of time, I suggest adding (t) next to the vector elements TRA

L216-218: I suggest revision of this statement. it's a product of a column vector by a row vector C(t), where the elements of the column vector are the fractions that were not transferred among the pools. ==Nice formulation!== TRA

L240: the focus of this publication is different, I think a more appropriate reference is this one: https://doi.org/10.5194/gmd-5-1259-2012 TRA

L278: N instead of n? TRA

L440-443: this statement does not align with the results of the NLS regression of the

respiration data performed in this study The NLS results were removed, but the difference between MCMC calibration with CO2 and with SOCCO2 or SOC was clear enough to allow reformulation of the statement. This was explained already in detail, when discussing the implication of the larger tail in Ricker function with MCMC CO2 data and can be summarized as below.

The main reason for lower SWCopt was the including SOC data or combination of SOC and CO2 data in model fitting, as the model fitting based only on CO2 showed larger SWCopt and larger tail (descending slope) of the Ricker moisture function.

L283: "the mean volumetric…" L783 TRA

Figure 4: please include legend for colors TRA

---

## Referee Report (RR1)

**General comments:**

The manuscript is actually in my opinion pretty good, it gives a very good number of details and the study and methods are described in depths. Some details are missing but overall, it is extensively described.

Significance is also very good, this is a rather important improvement of a model that is used quite a lot.

I do have some concerns though.

[partially required, could be discussed]

My main concern with the comparison between former Yasso07 version and yours is that yours was calibrated, the others if I understand well no. Ok, you calibrated only the scaling function for xi, but still the previous functions were calibrated on different data and might have hit another optimum on this particular dataset, and like this it becomes difficult to understand if the improvement in fitness is because of the structural improvement or because of the calibration. This might impact your Fig. 6 heavily.

I don't consider this a major flaw of the manuscript, since you are anyway declaring properly your methods and the reader can judge, but I would want to elaborate a bit in the discussion about this possible risk, giving some caution to the reader in interpreting your results.

Your results are reasonable. I don't see a reason why a monotonic moisture function could not be much worse than a non-monotonic (more specifically bitonic, even if does sound a bit cacophonic, I agree) one so I really believe the results, it contains important and much needed improvements for a broadly adopted model. But your comparison is at least quantitatively flawed if you affirm your structure was superior, since you cannot tell apart the effect of the structural change and the one of the calibrations. I advise caution here, your structure is superior also according to me but based mainly on inductive reasoning.

[required]

There is some inconsistency with how you refer to figure, sometimes Figure sometimes Fig. (in the text)

[required]

Density of the data: it is not immediate to understand exactly what the time series considered are, I mean how m any points over time each time series considered has. Are they same density or not (I guess not)? How are they spaced, evenly or not? When were the points collected, at what intervals? It is somehow possible to figure this out, but it is a crucial detail for understanding the calibration (the posterior likelihoods from your two objectives might have very different shapes if you compare a sparse time series with a very dense one, the sparse will have many peaks. IUt might also contribute to explain the discrepancies between the calibrations) and I think it should be a detail that stands out clearly in M&M.

[not required, just a suggestion]

I am honestly surprised to see how few models are utilizing a non-monotonic moisture reduction already, it's a decade-old discussion now and seems quite solved. Can you discuss specifically this topic more explicitly in the intro? Like which are the models which already updated the moisture reduction to non-monotonic? Are there some? A bit of state-of-the-art (like 2-3 lines, not more, classifying which models use monotonic and which bitonic if there are, and if there aren't then you can very rightfully claim a very big leap forward in terms of SOC model applicability).

[partially required, at least articulated answer appreciated]
When it comes to the optimum of your calibrated moisture scaling function, my guess is that it is different from other studies because of depth issues. You do not consider subsoil in your model, so you are working with assuming some mean water content, while water content will vary wildly in the profile. Even assuming the same depth of the water table, an organic soil will likely have a different depth/SWC curve than a mineral one, so it will regress to the mean with a different cumulative function. This issue could be discussed more (or other issues you believe could explain that discrepancy if you see others).
In your previous answer to the referees, you state that most respiration comes from the upper layer… I don't think this is necessarily true in an organic soil. In mineral soils this is true because most SOC is there, but an organic soil has a different SOC distribution, and when the water table gets lower than 30 cm you will have respiration also from those layers. I might of course be wrong but I would want to be shown wrong, in that case. Why do you think an organic soil with a low water table would have most of its respiration on 0-30? And how much is it "most", 60%? 95%?
It would be interesting to see the model residuals of your three calibrations (all of them, not just mean) plotted against SOC content, I personally expect them to follow some kind of pattern. I am not requiring this for the manuscript, but it would be something nice to see.

[not required, just a suggestion]
What is the implication on SOC stocks predictions of your three calibrations?
More extensively: it seems from your calibrations that SOC and $CO_2$ time series clearly contains information about different processes. For example, the $CO_2$ could contain information about hysteretic phenomena which are not all captured in the model, hence the two sources do not reconcile fully, as you already discuss.
In terms of applications, depending on the scope of the modeling I would choose one or the other unless the discrepancies are solved. If my aim is to simulate SOC stocks, I should be better off with the SOC only calibration, while if my aim is to simulate both I would accept a likely reduction in SOC fitness to get a better $CO_2$ representation.
It is possible to operate these choices based on table 2 anyway so this is not a required change, just making you aware I would reason this way if I had to apply the model.
A suggestion: plotting the posterior likelihoods of the two calibration objectives might help you understand more. Are they skewed, for example?

[partially required, advise caution to the readers]
Your conclusions are maybe too aggressive. You find a rather different optimum compared to many other studies, 14% to 27% seems quite low, and those studies were based on many data from lab (and also field I guess). There is something weird there, might be some missing processes (which I guess is missing depth), it would be dangerous to extrapolate before understanding what it is. If for example water table is involved, you risk doing wrong extrapolations when you change hydrology radically. Climate change extrapolations might not work so well if they change the hydrology of the sites (if hydrology was involved in the discrepancies between your results and the literature you cite). If you want to extrapolate such conclusions, I think you would need to discuss a bit more the discrepancies speculating some mechanisms, to then justify that the extrapolation is possible.
I mean, it is possible that what you affirm is true, but I would use some words of caution too.

**Specific comments:**

Line 40: Unimodal. I am not sure about this definition. It is true that an exponential or linear such as what was in Yasso before is not strictly unimodal, since it is strictly increasing and does not have any

distinct peak, so your definition might work. But this definition can be more relaxed, meaning that a function does not have multiple modes, so also the "old" function could be seen as unimodal. I would have personally referred to this concept as non-monotonic (or even better you can use, I think, the term "bitonic"), as opposed to the former monotonic function.
I mean, if you call your non-monotonic function "unimodal", how do you call then the function previously used? You propose a unimodal function instead of what? Could you define the two functions in a same phrase, this instead of that?
Just a suggestion.

Line 73: What do you mean with "a functional form reaching saturation"? That is monotonic, as in the opposite of (unimodal && multimodal)? Also your proposed function reaches saturation, it saturates at the optimum.

Line 78: modify "all kind" with something like "various", "a lot of" or similar, such absolute does not work in a scientific context (I get what you mean, though)

Line 84-86: Both statements are true but seem unrelated. One thing is that even if you calibrated a non-monotonic function like Moyano on mineral soils the same calibration won't probably work on organic soils, another is the fact that a monotonic function cannot represent the anoxic limitation process. It is hard to read and to get what you mean like this.

Line 95-96: if you describe this, then how was the functions improved in this study? Just scaling, or non-monotonic (with oxygen limitation processes)?

Line 102: … I wouldn't call Yasso particularly "parsimonious" in its class, compared with Century or RothC I mean, they are quite similar in terms of complexity, no? One could say Q is parsimonious, but Yasso should not have at all less parameters than RothC, right? It is a rather simple model class, though, I agree with that. No need to modify this for me, just be aware of how I read it.

Line 105: With SWC are you talking about the whole profile, total mm/m^2 kind of, or the gravimetric/volumetric water content (like g/g^1 or percent of pore space)? I think you should define SWC here to avoid ambiguity, there are many possible way to express it.

Line 108: What does "global" mean in this context? Meaning that the parameter values are considered constant everywhere? I ask because in some context you might be referring to a parameter space for example, as in "local and global optima".

Line 132-153: it is hard to understand what is the time resolution of these time series. How often did you measure, for each variable?

Line 153-168: Same. Was this one flux measurement each campaign, or more often?

Line197-199: Wait, do you mean that the Yasso07._xi_TW is not calibrated? I see now better what global means here, you mean the optimum of that specific model from previous calibrations on other datasets?

Line 285: "(ter Braak and Vrugt, 2008)" it's probably a typo

Line 374-375: I would say this phrase is redundant, nowadays Bayesian data assimilation approach has proven useless in countless applications. If you do not judge it redundant, why do you choose these specific studies over many others?

Line 387-390: Do you mean JULES has a constant reduction, not scaling with moisture?!? Just to be sure, I would have assumed these models were already much less rough on moisture reduction.

Line 475-490: I do not see any Arrhenius derived function. In Eq. 3 you have some kind of Q10 function. The Q10 function is quite rough, and it has indeed problems for very low and very high temperatures. See attached plot, where I plot only the lower end. You are using a function that is far from optimal at very low temperature, which in Finnish soils you are going to encounter often, so it's not surprising the model is not very good in those situations. Given the low respiration in those periods though the error should not be a very big issue, but I think you need to update your description if you did not use an Arrhenius or derived functions.

[Figure]

Line 503-504:; what do you mean that SOC stocks had the largest influence on moisture optimum? I guess you mean the opposite. Or do you mean that they were influencing the calibration the most? If so, from where do you derive this extrapolation?

Figure 2: there is probably something wrong in panel a), the smaller boxplots are not lining up. My guess is that you are not taking the right x points when you overlap the second plot (I guess you did this in Base R) as you seem to be doing in panel b)

Figure 2 caption: what kind of mean are you showing for SWC? Annual? Overall?

Figure 5: please describe more precisely the three dimensions you are showing here. What is on the Z axis? Is that the value of the resulting scaling xi_d?

In this case, which is how I interpret the plots, this plot is a bit redundant. It seems to show exactly the same data than Figure 4, since the two modifiers combine linearly, just in a slightly different way.

---

## Author Response (AR2)

Egusphere-2023-1523

**Reply to reviewers on "Modeling boreal forest's mineral soil and peat C dynamics with Yasso07 model coupled with Ricker moisture modifier"**

Boris Ťupek*, Aleksi Lehtonen, Alla Yurova, Rose Abramoff, Bertrand Guenet, Elisa Bruni, Samuli Launiainen, Mikko Peltoniemi, Shoji Hashimoto, Xianglin Tian, Juha Heikkinen, Kari Minkkinen, and Raisa Mäkipää (Note, Stefano Manzoni decided to not to be listed as co-author in the final version anymore due to too little contribution to warrant co-authorship. His contribution by comments on early version of the manuscript has been instead acknowledged). *boris.tupek@luke.fi

We thank both reviewers for thoughtful and insightful evaluation of our study, and for constructive comments which helped to improve the paper! Our replies are highlighted in yellow, or green when referring to the implementation of the comments in the revised paper.

**Largest revision of discussion in "4.1 The moisture response" relevant to main comments of both referees:**

**Revision related to comments on low moisture optimum:**

[L404-439] However, uncertainty in functional moisture - soil respiration dependencies are high (Sierra et al., 2015; Falloon et al., 2011) and dependencies vary with the soil properties, e.g., SWC optimum increases for soils with higher organic C content (from 30% to 75% SWC, Moyano et al., 2012, 2013). The $\xi_{AR}$ function's $SWC_{opt}$ found in dry and well-drained conditions and reduction of default decomposition rates (k) with increasing soil wetness contrasted with responses from the short-term laboratory incubation soil respiration studies (weeks, months) showing increase in decomposition from dry conditions until reduction in very wet (Sierra et al., 2017; Moyano et al., 2012, 2013; Kelly et al., 2000; Skopp et al., 1990; Yan et al., 2018). The $\xi_{AR}$ optimized with SOC and CO2 data showed that the optimum/maximum decomposition rate in the forest-mire ecotone was in dry well-drained conditions around 14% of mean long-term near surface SWC (around 20 % WFPS, corresponding to sub-xeric and mesic forest site types) ($SWC_{opt}$ parameters inferred from a parameter in Table 1, Fig. 4b) whereas the moisture optimum of studies based only on respiration from laboratory soil incubations was around 40% - 60% (Fairbairn et al., 2023; Moyano et al., 2013; Kelly et al., 2000; Skopp et al., 1990; Yan et al., 2018).

The moisture optimum derived from the field sites soil respiration datasets from a larger moisture range was found in 50% water-filled pore space (WFPS) and corresponding to around 31% SWC assuming mean porosity of 62%, Hashimoto et al., 2011). Our $SWC_{opt}$ between 14 and 27% SWC (Table 1) was comparable to the optimum derived from the field sites data which was lower compared to laboratory incubations. The $SWC_{opt}$ discrepancy of the $\xi_{AR}$ function highlights the difference between (1) the responses from the field-based or long-term soil respiration measurements reflecting moisture responses of older, stabilized and slowly decomposing SOC, and (2) the short-term incubation-based soil respiration studies which predominantly capture decomposition of newly available, labile and rapidly decomposing, SOC pool (González-Domínguez et al., 2022; Huang and Hall, 2017). Over longer periods of incubation high Q10 can be observed (Zhou et al., 2019). The enhanced C mineralization can occur during periods of elevated moisture under Fe reduction when microbes can access previously protected labile C (Huang and Hall, 2017). The incubations are short term (from few days to few months) and are useful to identify short term processes. Moreover, they are performed on disturbed soils (sometimes even sieved) and therefore the soil structure is not representative of the field.

The ecosystem scale application of moisture reduction functions obtained in the laboratory can be hindered by several factors. There are number of feedback mechanisms which modify the response obtained on a limited size soil sample. Among them is a change in microbial community composition, the texture-and- structure-dependent effect of pore-scale connectedness of soil solutions and competition between plants and microorganism for resources under different environmental stress conditions. Under changing climate these feedback mechanisms may lead

to the system behavior unpredictable from extrapolation. Therefore, the validation of the models at the site level with series of various in-situ stress levels is necessary for reliable future predictions.

**Revision related to comments on model calibration and further development of model structure:**

[L467-471] Including SOC data or combination of SOC and CO2 data in model fitting resulted to lower SWCopt, and the model fitting based only on CO2 showed larger SWCopt and larger tail (descending slope) of the Ricker moisture function. Thus, in comparison to other studies, which dependencies were limited to relatively short-term responses of only soil heterotrophic CO2 respiration from mainly mineral soils in laboratory conditions, the differences in SWCopt observed in our studies could be expected from difference in data source used in model calibration.

[L495 -509] In this study, we constrained the soil carbon model using both SOC (stock) and CO2 (flow) data. Few studies have constrained the soil carbon model to both SOC and CO2 data. Our study demonstrates the importance of extensive constraints on the soil carbon model to obtain a reliable model output. The SOC constraint improved the model performance; at the same time, intensive SOC and CO2 constraint did not result in the improvement of model performance, which implies the need for further model development and testing. One potential improvement in modelling could be the different responses to the environment (e.g., soil moisture) among different pools like the temperature dependency separated between the soil layers and soil C fractions in more recent versions of Yasso model e.g., Yasso15 and Yasso20 (Viskari et al. 2020, 2022). The Yasso07 model adapts one common response function among different pools for simplicity; however, the fresh plant litter moisture limitation of decomposition may be expected to differ from the moisture limitation on older stabilized C in the humus horizon and mineral-associated C. Another factor could be the vertical process. SOC is vertically distributed in the soil, and soil C fractions differ among soil depths. Accounting for the depth of the soil layer with the largest proportion of net CO2 emissions (Davidson et al. 2006, 2012) which is expected to vary with fluctuating water level in forested peatlands may further improve the soil respiration estimates for organic soils. On a process level the key to understanding of the difference in moisture reduction function at different soil depth may lay in the nature of the physical and biochemical availability of substrate to enzymes released by microbial decomposers (Sainte-Marie et al, 2021).

References added to the discussion:
    Davidson, E.A., Savage, K.E., Trumbore, S.E. and Borken, W., 2006. Vertical partitioning of CO2 production within a temperate forest soil. Global Change Biology, 12, 944-956. https://doi.org/10.1111/j.1365-2486.2005.01142.x
    Sainte-Marie, J., Barrandon, M., Saint-André, L., Gelhaye, E., Martin, F., Derrien, D., 2021. C-STABILITY an innovative modeling framework to leverage the continuous representation of organic matter. Nat Commun 12, 810. https://doi.org/10.1038/s41467-021-21079-6
    Viskari, T., Pusa, J., Fer, I., Repo, A., Vira, J., and Liski, J. 2022 Calibrating the soil organic carbon model Yasso20 with multiple datasets, Geosci. Model Dev., 15, 1735–1752. https://doi.org/10.5194/gmd-15-1735-2022
    Viskari, T., Laine, M., Kulmala, L., Mäkelä, J., Fer, I., and Liski, J. 2020. Improving Yasso15 soil carbon model estimates with ensemble adjustment Kalman filter state data assimilation, Geosci. Model Dev., 13, 5959–5971. https://doi.org/10.5194/gmd-13-5959-2020

**Detailed replies to specific comments and their improvements:**

**Referee#1 Dr. Lorenzo Menichetti  #############**

**General comments:**
The manuscript is actually in my opinion pretty good, it gives a very good number of details and the study and methods are described in depths. Some details are missing but overall, it is extensively described.
Significance is also very good, this is a rather important improvement of a model that is used quite a lot.
Thank you!

I do have some concerns though.

[partially required, could be discussed]
My main concern with the comparison between former Yasso07 version and yours is that yours was calibrated, the others if I understand well no. Ok, you calibrated only the scaling function for xi, but still the previous functions were calibrated on different data and might have hit another optimum on this particular dataset, and like this it becomes difficult to understand if the improvement in fitness is because of the structural improvement or because of the calibration. This might impact your Fig. 6 heavily.
I don't consider this a major flaw of the manuscript, since you are anyway declaring properly your methods and the reader can judge, but I would want to elaborate a bit in the discussion about this possible risk, giving some caution to the reader in interpreting your results.
Your results are reasonable. I don't see a reason why a monotonic moisture function could not be much worse than a non-monotonic (more specifically bitonic, even if does sound a bit cacophonic, I agree) one so I really believe the results, it contains important and much needed improvements for a broadly adopted model. But your comparison is at least quantitatively flawed if you affirm your structure was superior, since you cannot tell apart the effect of the structural change and the one of the calibrations. I advise caution here, your structure is superior also according to me but based mainly on inductive reasoning.

We revised discussion.
[L384-389] The original Yasso07 monotonic precipitation function is effective due to easily available data on upper boundary condition, but also flawed in case of shallow water table when the lower boundary is equally important in defining the water content on the soil. Therefore, the usage of soil water content as a variable is structurally superior, and can be proved by inductive reasoning, e.g., from the test model runs. Separating the effect of structure against calibration would require more test runs with data from larger number of study sites.

[required]
There is some inconsistency with how you refer to figure, sometimes Figure sometimes Fig. (in the text) The "Figure" [L157] was corrected to "Fig." as on other instances in the text. Otherwise in figure captions we use "Figure" according to the manuscript preparation guidelines.

[required]
Density of the data: it is not immediate to understand exactly what the time series considered are, I mean how m any points over time each time series considered has. Are they same

density or not (I guess not)? How are they spaced, evenly or not? When were the points collected, at what intervals? It is somehow possible to figure this out, but it is a crucial detail for understanding the calibration (the posterior likelihoods from your two objectives might have very different shapes if you compare a sparse time series with a very dense one, the sparse will have many peaks. IUt might also contribute to explain the discrepancies between the calibrations) and I think it should be a detail that stands out clearly in M&M.
We revised discussion.
[L510-514] The less frequent measurements during the near zero soil temperature might have affected the fit of the temperature function. However, our main emphasis was on the moisture which in near zero temperature conditions plays only a minor role on controlling respiration.

Density of $CO_2$, temperature and SWC measurements can be seen in Fig.3.

[not required, just a suggestion]
I am honestly surprised to see how few models are utilizing a non-monotonic moisture reduction already, it's a decade-old discussion now and seems quite solved. Can you discuss specifically this topic more explicitly in the intro? Like which are the models which already updated the moisture reduction to non-monotonic? Are there some? A bit of state-of-the-art (like 2-3 lines, not more, classifying which models use monotonic and which bitonic if there are, and if there aren't then you can very rightfully claim a very big leap forward in terms of SOC model applicability).
We clarified in introduction [L84-86] which models, use non-monotonic and monotonic functions.

[partially required, at least articulated answer appreciated]
When it comes to the optimum of your calibrated moisture scaling function, my guess is that it is different from other studies because of depth issues. You do not consider subsoil in your model, so you are working with assuming some mean water content, while water content will vary wildly in the profile. Even assuming the same depth of the water table, an organic soil will likely have a different depth/SWC curve than a mineral one, so it will regress to the mean with a different cumulative function. This issue could be discussed more (or other issues you believe could explain that discrepancy if you see others).
In your previous answer to the referees, you state that most respiration comes from the upper layer… I don't think this is necessarily true in an organic soil. In mineral soils this is true because most SOC is there, but an organic soil has a different SOC distribution, and when the water table gets lower than 30 cm you will have respiration also from those layers. I might of course be wrong but I would want to be shown wrong, in that case. Why do you think an organic soil with a low water table would have most of its respiration on 0-30? And how much is it "most", 60%? 95%?
It would be interesting to see the model residuals of your three calibrations (all of them, not just mean) plotted against SOC content, I personally expect them to follow some kind of pattern. I am not requiring this for the manuscript, but it would be something nice to see.

revised discussion with added text:
[L405-444] text was relevant for both referees and it was listed in the main revision

[not required, just a suggestion]
What is the implication on SOC stocks predictions of your three calibrations?
More extensively: it seems from your calibrations that SOC and $CO_2$ time series clearly contains information about different processes. For example, the $CO_2$ could contain information about

hysteretic phenomena which are not all captured in the model, hence the two sources do not reconcile fully, as you already discuss.

In terms of applications, depending on the scope of the modeling I would choose one or the other unless the discrepancies are solved. If my aim is to simulate SOC stocks, I should be better off with the SOC only calibration, while if my aim is to simulate both I would accept a likely reduction in SOC fitness to get a better CO2 representation.

It is possible to operate these choices based on table 2 anyway so this is not a required change, just making you aware I would reason this way if I had to apply the model.

A suggestion: plotting the posterior likelihoods of the two calibration objectives might help you understand more. Are they skewed, for example?

text added into discussion.

[423-426] In its impact on decomposition of the $\xi_{AR}$ functions (calibrated with SOC, SOCCO2, and CO2 data) incorporated into Yasso07 soil C model were comparable (e.g., all found the moisture optimum in well-drained soils of forest-mire ecotone). Although, the soil temperature and moisture functions showed a relatively small differences in Q10 between the model fits, the "a" parameter of the moisture functions of CO2 based fit was larger than from SOC and SOCCO2 fit (Table 1).

[partially required, advise caution to the readers]
Your conclusions are maybe too aggressive. You find a rather different optimum compared to many other studies, 14% to 27% seems quite low, and those studies were based on many data from lab (and also field I guess). There is something weird there, might be some missing processes (which I guess is missing depth), it would be dangerous to extrapolate before understanding what it is. If for example water table is involved, you risk doing wrong extrapolations when you change hydrology radically. Climate change extrapolations might not work so well if they change the hydrology of the sites (if hydrology was involved in the discrepancies between your results and the literature you cite). If you want to extrapolate such conclusions, I think you would need to discuss a bit more the discrepancies speculating some mechanisms, to then justify that the extrapolation is possible.

I mean, it is possible that what you affirm is true, but I would use some words of caution too.

in addition to previous mentioned revised discussion [L405-423] we also revised conclusions:
adding text:
[L517-518] In this study we emphasized on improving representation of the response of soil organic C stock change and respiration to soil moisture in Yasso07 model for selected forest- mire ecosystems.
reformulating:
[L533-535] Also, the non-monotonic Ricker function with a moisture optimum in well-drained mineral soils needs further evaluation with regional boreal forest data.
adding text:
[L534-537] The exact representation of the functional form of the soil moisture dependency is considered characteristic to conditions of our study e.g., the distribution of organic and mineral soil forests in the data. Broader extrapolation of the conclusions e.g., to climate change or forest management on drained peatlands would require more model testing with spatially larger data and lower water levels in forests on organic soils.
reformulating:
[L537-542] However, if the soil moisture optimum of litter decomposition in forests on well drained mineral soils of boreal landscape proves to be robust, then in the future warmer and drier climates the boreal forest could be expected to enhance soil C emissions to the atmosphere due to water level drawdown of presently water-saturated peat soils with large C stocks. In

contrary, rewetting of previously drained peatlands could be expected to reduce soil C emissions, turning SOC loss to long-term C sequestration.

**Specific comments:**

Line 40: Unimodal. I am not sure about this definition. It is true that an exponential or linear such as what was in Yasso before is not strictly unimodal, since it is strictly increasing and does not have any distinct peak, so your definition might work. But this definition can be more relaxed, meaning that a function does not have multiple modes, so also the "old" function could be seen as unimodal. I would have personally referred to this concept as non-monotonic (or even better you can use, I think, the term "bitonic"), as opposed to the former monotonic function.

I mean, if you call your non-monotonic function "unimodal", how do you call then the function previously used? You propose a unimodal function instead of what? Could you define the two functions in a same phrase, this instead of that?

Just a suggestion.

revised:

[L36-42] … we revised the original precipitation-based monotonic saturation dependency of the Yasso07 soil carbon model by using non-monotonic Ricker function based on soil volumetric water content. We fit the revised functional dependency of moisture to the observed microbial respiration and SOC and compared its performance against the original Yasso07 model and the version used in the JSBACH land surface model with a reduction constant for decomposition rates in wetlands.

The Yasso07 soil C model coupled with the calibrated unimodal Ricker moisture function with an optimum in well drained…

Line 73: What do you mean with "a functional form reaching saturation"? That is monotonic, as in the opposite of (unimodal && multimodal)? Also your proposed function reaches saturation, it saturates at the optimum.

revised:

[L75-76] For example, the moisture decomposition dependency in the Yasso07 soil C model (Tuomi et al., 2011, 2009) is based on annual precipitation and has a form of monotonic saturation and is uninformed about soil characteristics.

added                                                                                                                    text:

[L76-77] By a monotonic saturation function, we mean a function which is entirely nondecreasing, initially increasing rapidly and later slowly approaching maximum.

Line 78: modify "all kind" with something like "various", "a lot of" or similar, such absolute does not work in a scientific context (I get what you mean, though) revised: various

Line 84-86: Both statements are true but seem unrelated. One thing is that even if you calibrated a non-monotonic function like Moyano on mineral soils the same calibration won't probably work on organic soils, another is the fact that a monotonic function cannot represent the anoxic limitation process. It is hard to read and to get what you mean like this.

reformulated:

[L93 – L94] However, the inhibition of decomposition can be accounted for even in monotonic functions, e.g.by adding a reduction parameter such as "anerb" in CENTURY.

Line 95-96: if you describe this, then how was the functions improved in this study? Just scaling, or non-monotonic (with oxygen limitation processes)?

revised: [L102] with the anoxic inhibition

Line 102: … I wouldn't call Yasso particularly "parsimonious" in its class, compared with Century or RothC I mean, they are quite similar in terms of complexity, no? One could say Q is parsimonious, but Yasso should not have at all less parameters than RothC, right? It is a rather simple model class, though, I agree with that. No need to modify this for me, just be aware of how I read it. Yes, RothC and Century soil C sub-module are in the same class as Yasso.

Line 105: With SWC are you talking about the whole profile, total mm/m^2 kind of, or the gravimetric/volumetric water content (like g/g^1 or percent of pore space)? I think you should define SWC here to avoid ambiguity, there are many possible way to express it.
revised: [L106] soil volumetric water content

Line 108: What does "global" mean in this context? Meaning that the parameter values are considered constant everywhere? I ask because in some context you might be referring to a parameter space for example, as in "local and global optima".
revised: global was deleted, it means calibration with global litter decomposition dataset

Line 132-153: it is hard to understand what is the time resolution of these time series. How often did you measure, for each variable?
revised:
[L140-143] The measurement campaigns were conducted weekly, and we measured each plot once and all plots in one or two days between 7 am and 6 pm during the vegetative season of 2004 (July-November), 2005 (May-November), 2006 (May-September), and monthly during the non-vegetative season (December-April).

Line 153-168: Same. Was this one flux measurement each campaign, or more often?
see the answer above

Line197-199: Wait, do you mean that the Yasso07._xi_TW is not calibrated? I see now better what global means here, you mean the optimum of that specific model from previous calibrations on other datasets? the calibration of Yasso07.$\xi$TW is explained in L236-240
Line 285: "(ter Braak and Vrugt, 2008)" it's probably a typo
revised: deleted

Line 374-375: I would say this phrase is redundant, nowadays Bayesian data assimilation approach has proven useless in countless applications. If you do not judge it redundant, why do you choose these specific studies over many others?
we moved the sentence to methods and revised it:
[L271-272] The Bayesian MCMC data assimilation has proven useful in improving soil organic carbon estimates (e.g., Xu et al., 2006; Hararuk et al., 2014).

Line 387-390: Do you mean JULES has a constant reduction, not scaling with moisture?!? Just to be sure, I would have assumed these models were already much less rough on moisture reduction.

revised:
[396-398] The 96% reduction is comparable to JULES which accounts for oxygen inhibition with gradual reduction of decomposition from the maximum rate 1 at the moisture optimum (30% - 75% SWC) to a reduced rate 0.2 in water-saturated peat soils (Chadburn et al., 2022).

Line 475-490: I do not see any Arrhenius derived function. In Eq. 3 you have some kind of Q10

function. The Q10 function is quite rough, and it has indeed problems for very low and very high temperatures. See attached plot, where I plot only the lower end. You are using a function that is far from optimal at very low temperature, which in Finnish soils you are going to encounter often, so it's not surprising the model is not very good in those situations. Given the low respiration in those periods though the error should not be a very big issue, but I think you need to update your description if you did not use an Arrhenius or derived functions.

revised:

[L221-222] … we re-defined the $\xi(tm)$ function for the use with soil temperature based on a Q10 exponential function to $T_5$ (used by Davidson et al. (2012) as an alternative to Arrhenius kinetics) …

later on, we refer to it as "Q10 function"

[Figure]

Line 503-504:; what do you mean that SOC stocks had the largest influence on moisture optimum? I guess you mean the opposite. Or do you mean that they were influencing the calibration the most? If so, from where do you derive this extrapolation?

revised:

[521-523] SOC stocks had the largest influence on calibrated the moisture optimum, when they were included along with fluxes in optimization. This could be inferred from the same calibrated moisture optimum when using calibration with only SOC or SOCCO2 as data source, whereas for only CO2 based calibration the optimum differs.

Figure 2: there is probably something wrong in panel a), the smaller boxplots are not lining up. My guess is that you are not taking the right x points when you overlap the second plot (I guess you did this in Base R) as you seem to be doing in panel b)

small misalignment is intentional as it helps to distinguishing the error lines

Figure 2 caption: what kind of mean are you showing for SWC? Annual? Overall?
revised:
[L816] of all measured values

Figure 5: please describe more precisely the three dimensions you are showing here. What is on the Z axis? Is that the value of the resulting scaling xi_d?
revised:
[L836] The colors and contour lines showing optimized environmental modifier of default decomposition rates $\xi$

In this case, which is how I interpret the plots, this plot is a bit redundant. It seems to show exactly the same data than Figure 4, since the two modifiers combine linearly, just in a slightly different way.
text added in results:
[L343-344] The combined non-linear temperature and moisture response in whole climate data range showed larger nonlinear variation of the change in $\xi$ for mineral soil forests than forest mire transitions and peatlands (Fig. 5).
revised:
[L345-347] The $\xi_{AR}$ in the Fig. 5 panels a and b are similar showing that both SOC and SOCCO2 parameterization is almost the same whereas the $\xi_{AR}$ in the Fig.5c $\xi_{AR}$ is different.

**Referee#2 ##################**
I think the revised manuscript is greatly improved, and should be published subject to minor revisions.
Thank you!

In the revised manuscript assimilation of both CO2 and SOC data resulted in worsened performance of Yasso in simulating CO2 fluxes and SOC stocks compared to when only CO2 fluxes or only SOC stocks were assimilated. I.e. when CO2 fluxes alone were assimilated model performed better at simulating CO2 fluxes that when both SOC and CO2 fluxes were assimilated; similarly, when SOC stocks alone were assimilated model performed better at simulating SOC stocks than when both CO2 fluxes and SOC stocks were used for parameter calibration. To me this seems to point to the issues with model structure, because the model could not represent two observation types "in the best way" after calibration.
Not all questions [below] may be answered with the data you have, but if some might, it would be worth discussing.

In the interest of furthering model development, could you please include a paragraph in the discussion section with your thoughts about why the model performance worsens when two data types are assimilated and how the model structure could be improved based on the model-data mismatch patterns at your plots?

Should environmental limitation function be different for different pools?

How could incorporation of vertical dimension affect model performance?

we revised discussion:

[L495 -509] In this study, we constrained the soil carbon model using both SOC (stock) and CO2 (flow) data. Few studies have constrained the soil carbon model to both SOC and CO2 data. Our study demonstrates the importance of extensive constraints on the soil carbon model to obtain a reliable model output. The SOC constraint improved the model performance; at the same time, intensive CO2 constraint did not result in the improvement of model performance, which implies the need for further model testing and improvements. One potential improvement in modelling could be the different responses to the environment (e.g., soil moisture) among different pools like the temperature dependency separated between the soil layers and soil C fractions in more recent versions of Yasso model e.g., Yasso15 and Yasso20 (Viskari et al. 2020, 2022). The Yasso07 model adapts one common response function among different pools for simplicity; however, the fresh plant litter moisture limitation of decomposition may be expected to differ from the moisture limitation on older stabilized C in the humus horizon and mineral-associated C. Another factor could be the vertical process. SOC is vertically distributed in the soil, and soil C fractions differ among soil depths. Accounting for the depth of the soil layer with the largest proportion of net CO2 emissions (Davidson et al. 2006, 2012) which is expected to vary with fluctuating water level in forested peatlands may further improve the soil respiration estimates for organic soils. On a process level the key to understanding of the difference in moisture reduction function at different soil depth may lay in the nature of the physical and biochemical availability of substrate to enzymes released by microbial decomposers (Sainte-Marie et al, 2021).

Davidson, E.A., Savage, K.E., Trumbore, S.E. and Borken, W., 2006. Vertical partitioning of CO2 production within a temperate forest soil. Global Change Biology, 12, 944-956. https://doi.org/10.1111/j.1365-2486.2005.01142.x

Sainte-Marie, J., Barrandon, M., Saint-André, L., Gelhaye, E., Martin, F., Derrien, D., 2021. C-STABILITY an innovative modeling framework to leverage the continuous representation of organic matter. Nat Commun 12, 810. https://doi.org/10.1038/s41467-021-21079-6

Viskari, T., Pusa, J., Fer, I., Repo, A., Vira, J., and Liski, J. 2022 Calibrating the soil organic carbon model Yasso20 with multiple datasets, Geosci. Model Dev., 15, 1735–1752. https://doi.org/10.5194/gmd-15-1735-2022

Viskari, T., Laine, M., Kulmala, L., Mäkelä, J., Fer, I., and Liski, J. 2020. Improving Yasso15 soil carbon model estimates with ensemble adjustment Kalman filter state data assimilation, Geosci. Model Dev., 13, 5959–5971. https://doi.org/10.5194/gmd-13-5959-2020